# Immune Checkpoints Pathways in Head and Neck Squamous Cell Carcinoma

**DOI:** 10.3390/cancers13051018

**Published:** 2021-03-01

**Authors:** Florencia Veigas, Yamil D. Mahmoud, Joaquin Merlo, Adriana Rinflerch, Gabriel Adrian Rabinovich, María Romina Girotti

**Affiliations:** 1Laboratorio de Inmuno Oncología Traslacional, Instituto of Biología y Medicina Experimental, Consejo Nacional de Investigaciones Científicas y Técnicas (CONICET), Buenos Aires C1428ADN, Argentina; fveigas@dna.uba.ar (F.V.); ymahmoud@dna.uba.ar (Y.D.M.); jmerlo@dna.uba.ar (J.M.); 2Laboratorio GIGA, Facultad de Ciencias Exactas, Químicas y Naturales, Instituto de Biología Subtropical, Universidad Nacional de Misiones, CONICET, Posadas N3300NFK, Misiones, Argentina; adriana.rinflerch@hospitalitaliano.edu.ar; 3Laboratorio de Inmunopatología, Instituto of Biología y Medicina Experimental, CONICET, Buenos Aires C1428ADN, Argentina; gabriel.r@ibyme.conicet.gov.ar; 4Facultad de Ciencias Exactas y Naturales, Universidad de Buenos Aires, Buenos Aires C1428EGA, Argentina

**Keywords:** immune infiltrate, head and neck cancer, immunotherapy, immune checkpoint

## Abstract

**Simple Summary:**

During the last decades, scientific advances in immuno-oncology and a better understanding of tumors’ immune profile led to the development of novel immunotherapeutic strategies, especially immune checkpoint inhibitors. The blockade of PD-1 by monoclonal antibodies (mAbs) is the only immunotherapy based on immune checkpoint pathways approved for head and neck squamous cell carcinoma. As only a small fraction of patients perceives clinical benefit, understanding the molecular mechanisms and signaling pathways activated by the immune checkpoints and other tumor intrinsic features that modulate the immune infiltrate is crucial to better select patients for immunotherapy treatment and to develop novel therapeutic strategies. We here review the immune escape mechanisms of head and neck tumors, with a particular focus on the immune checkpoints, their role as therapeutic targets, and the predictive biomarkers of response to anti-PD-1/PD-L1 therapy. We also summarize the ongoing clinical trials testing several combinations of immune checkpoint inhibitors with other therapeutic approaches to improve patient outcomes.

**Abstract:**

Head and neck squamous cell carcinoma (HNSCC) is a heterogeneous group of tumors usually diagnosed at an advanced stage and characterized by a poor prognosis. The main risk factors associated with its development include tobacco and alcohol consumption and Human Papillomavirus (HPV) infections. The immune system has a significant role in the oncogenesis and evolution of this cancer type. Notably, the immunosuppressive tumor microenvironment triggers immune escape through several mechanisms. The improved understanding of the antitumor immune response in solid tumors and the role of the immune checkpoint molecules and other immune regulators have led to the development of novel therapeutic strategies that revolutionized the clinical management of HNSCC. However, the limited overall response rate to immunotherapy urges identifying predictive biomarkers of response and resistance to treatment. Here, we review the role of the immune system and immune checkpoint pathways in HNSCC, the most relevant clinical findings linked to immunotherapeutic strategies and predictive biomarkers of response and future treatment perspectives.

## 1. Introduction

Head and neck cancers comprise a heterogeneous group of tumors arising in the head and neck region, with head and neck squamous cell carcinoma being the most frequent histology type. Particularly, HNSCC develops in the mucosal surfaces lining the paranasal sinuses, nasal and oral cavity, oropharynx, larynx, and hypopharynx [1]. HNSCC is the sixth most common cancer worldwide, and its incidence continues to rise [2]. By 2020, the number of new cases and deaths estimated worldwide was more than 800,000 and 400,000, respectively [3]. 

The main risk factors for developing HNSCC are tobacco smoke, alcohol consumption, and HPV infections [4,5,6]. HPV-positive (HPV^+^) cancers often arise in the oropharynx cavity and represent a different biological and clinical entity with a distinct mutation landscape characterized by markedly improved survival [7,8]. The genomic landscape of HNSCC is characterized by tumor heterogeneity and loss-of-function mutations in tumor-suppressor genes such as TP53 and FAT1, activation of oncogenes such as EGFR and PIK3CA, or inactivation by heterozygous and homozygous losses in CDKN2A (reviewed in Reference [8]). Interestingly, TP53 and CDKN2A, most frequently altered in HNSCC, are unaffected in HPV^+^ tumors, which are characterized by molecular alterations in the PI3K pathway [8]. 

The initial diagnosis of HNSCC is often clinical and is based on the patient’s symptoms that vary from chronic pain in the throat, tongue, or mouth to hoarseness or difficulty swallowing. Clinical examination of the head and neck is a major for HNSCC diagnosis as these patients could present non-healing ulcers, red/white patches in the mouth, or an irregular mass in the neck. Pathological confirmation of the diagnosis by analysis of a tumor biopsy is mandatory [1,9]. The microscopic appearance may vary as a function of tumor differentiation, and the main neoplastic alteration includes abnormal cellular organization, increased mitotic activity, and nuclear enlargement with pleomorphism [10]. The increasing incidence of HPV-associated HNSCC and its relevant role as a prognostic biomarker in oropharyngeal squamous cell carcinomas (OSCC) supports the molecular testing for HPV infection in all patients diagnosed with this HNSCC subtype. Current guidelines indicate the routine pathological evaluation of HPV in (1) newly diagnosed OSCC patients from the primary tumor or cervical nodal metastases by p16 immunohistochemistry (IHC), any additional HPV testing should be performed at the discretion of the pathologist or treating physician, and (2) patients with metastatic unknown primary from cervical upper- or mid-jugular chain lymph node by p16 IHC and then for p16^+^ tumors, HPV-specific detection assays must be done to confirm HPV status [11].

HNSCC subtypes are clinically, histologically, and molecularly distinct, and their treatment is mainly guided by the specific site of disease, size, age, tumor accessibility, performance status, and stage and is not based on genomic alterations or gene expression profiles. The NCCN and ESMO guidelines report that patients with early-stage tumors are recommended to be treated by a single arm-modality with either surgery or radiotherapy (RT), often providing similar survival rates or locoregional control. However, each case should be evaluated individually, as one modality may be better than the other under certain conditions. Particularly, the treatment is uniform within each HNSCC entity, being surgery usually preferred for oral cavity and paranasal sinus, and RT for nasopharyngeal carcinomas. Nevertheless, the choice of treatment modality should be based on the assessment of functional outcomes and perceived relative morbidity for each patient as well as institutional and patient preferences and experience [9,12]. Usually, early-stage cancers have a favorable prognosis with high cure rates, and the treatment improves long-term survival. However, patients with locally advanced HNSCC cancer carry a high risk of recurrence, distant metastases, and poor prognosis [1]. For locoregional advanced HNSCC, standard treatment often includes surgery plus adjuvant RT, or chemoradiotherapy (CRT), or primary CRT alone. Generally, surgery is preferred for the oral cavity followed by RT or CRT, while for other sites, surgery is reserved for smaller accessible primary tumors or for those whose response to induction CRT was poor. Particularly, concomitant CRT resulted in greater locoregional control and improved OS than RT alone, irrespective of the tumor location. Nonetheless, treatment decisions need to be individualized and carefully evaluated by a multispecialty team in a tumor board and several aspects should be considered: morbidity, toxic effects, and preservation of organ function, among others [9,12]. However, around 50% of patients with locally advanced HNSCC develop recurrence after primary treatment with metastatic, local, or regional disease and have a poor prognosis with a median overall survival (OS) under 12 months (six to nine months without treatment) [1]. Treatment of these patients is complex and requires the evaluation of tumor- and patient-related factors for selecting the optimal combination and sequencing of treatments. Patients with locoregional recurrence may be candidates for surgical salvage, and those patients not amenable to surgery are candidates for curative-intent RT or re-irradiation with or without chemotherapy depending on the time since prior treatment and related toxicity. On the other hand, for patients with recurrent or metastatic (R/M) HNSCC not amenable to the mentioned curative options, palliative systemic therapy is indicated [13]. In this regard, the standard of care (SoC) in the first-line setting from 2008 to 2019 has been the EXTREME regimen, consisting of platinum-based doublet chemotherapy with cetuximab and 5-fluorouracil while for second-line treatment until 2016 has been cetuximab, methotrexate, and taxane [14]. 

Over the last few years, a revolution started with the development and approval of immunomodulatory therapies, particularly Immune Checkpoint Blockade (ICB) therapies, for patients with R/M HNSCC. In 2020, the current FDA indications for Pembrolizumab treatment in HNSCC were as follows: (1) in combination with platinum and 5-fluorouracil for the first-line treatment of patients with metastatic or with unresectable, recurrent HNSCC; (2) as a single agent for the first-line treatment of patients with metastatic or with unresectable, recurrent HNSCC whose tumors express Programmed Death-Ligand 1 (PD-L1) (Combined Positive Score (CPS): ≥1) as determined by an FDA-approved test; (3) as a single agent for the treatment of patients with R/M HNSCC with disease progression on or after platinum-containing chemotherapy [15]. As with other tumor types, molecular biomarkers are critical for diagnosis, monitoring disease progression, and predicting response to treatment. Currently, only PD-L1 expression is widely used as a predictive biomarker for response to immune checkpoint inhibitors in HNSCC [16]. Indeed, current guidelines recommend the evaluation of PD-L1 in HNSCC tumors by IHC. Concerning HPV status, despite its prognostic value in OSCC patients, it does not change the treatment algorithm [9,12]. However, clinical differences between HPV^+^ and HPV^-^ patients are now being tested separately in several ongoing or just finished clinical trials to evaluate biological and treatment-related questions such as how HPV status impacts tumor response to treatment and de-escalation of the standard therapy for HPV^+^ tumors [17,18]. De-escalation aims to decrease toxicity and morbidity resulted from SoC, while maintaining tumor control, quality of life, and favorable survival [18]. Likewise, EGFR overexpression and specific TP53 in HNSCC are associated with poor survival outcomes but do not impact treatment algorithms [14,19]. Many biomarkers have been suggested to impact the diagnosis and prognosis of HNSCC, significantly. However, they still lack high specificity and sensitivity, low cost, high positive predictive value, clinical relevance, and short turnaround time [20,21].

This review revises the immune escape mechanisms in HNSCC, focusing on the immune checkpoint pathways and their role as therapeutic targets. Furthermore, we discuss the use of Immune Checkpoint Blockade (ICB) in HNSCC treatment, the ongoing clinical trials testing several combinations of immune checkpoint inhibitors with other therapeutic approaches, and novel predictive biomarkers of response to anti-PD-1/PD-L1 therapy.

## 2. Mechanisms of Immune Escape in HNSCC

HNSCC is considered an immunogenic tumor as it is often accompanied by a prominent immune infiltrate [22]. However, this infiltrate leads to impaired tumor recognition and elimination, given its immunosuppressive nature [23]. Tumor-immune escape, orchestrated by cellular and molecular regulatory networks, represents a key hallmark of cancer, generating a microenvironment permissive for tumor survival, progression, and treatment failure [24]. Understanding the mechanisms implicated in immune evasion is crucial to better select patients for immunotherapy treatment and develop novel therapeutic strategies. The main mechanisms underlying immune escape in HNSCC are alterations in the antigen presentation machinery and the composition and activation profile of tumor-infiltrating immune cells shifting the tumor microenvironment (TME) towards more permissive for tumor progression (reviewed in Reference [23]).

### 2.1. Composition and Activation Profile of Immune Cells in the TME

The TME is a complex network composed of different cell types and soluble factors like chemokines, cytokines, and growth factors. Immune cells, including CD4^+^ and CD8^+^ T cells, regulatory T cells (Tregs), tumor-associated macrophages (TAMs), and myeloid-derived suppressor cells (MDSCs) are the main immune cells frequently infiltrating tumors [25], including HNSCC.

Several studies have shown that T cells of HNSCC patients are poorly responsive to antigenic stimuli, have an altered cytokine profile, show low cytotoxic activity, and have impaired signaling via T cell receptor (TCR), favoring immune-escape [26,27,28,29]. In HNSCC patients, there is also a decrease in the absolute T cell count in both tumor and circulation, an effect possibly related to spontaneous apoptosis via Fas/FasL pathway [30]. A systematic review and meta-analysis proposed that the infiltration of CD8^+^ T cells has a favorable prognostic value in HNSCC patients, while CD4^+^ T cells’ role remains questionable [31]. Moreover, Tregs contribute to immunosuppression in HNSCC by secreting IL-10 and TGF-β1 to the TME, hindering the antitumor activity of effector T cells (Teff) [32,33,34]. Although Tregs in the TME and circulation of HNSCC patients overexpress immune checkpoints on their surface, infiltrating Tregs are more immunosuppressive [35]. Depending on the cancer type, Tregs have been associated with either poor or good clinical outcomes. In particular, for HNSCC is still controversial, and evidence in the literature supports the notion that Tregs are associated with enhanced progression-free survival (PFS) and OS [31,36,37,38]. 

MDSCs and TAMs stand out as the most immunosuppressive tumor-infiltrating cells. On the one hand, MDSCs dampen immune responses by inhibiting Teff cells’ functions, fostering Tregs, and polarizing macrophages towards a pro-tumoral phenotype in the TME. In HNSCC patients, an inflammatory infiltrate with a predominant granulocytic MDSCs profile is associated with poorer survival [39]. Moreover, a recent study demonstrated that MDSCs infiltrating these tumors exhibit greater immunosuppressive capacity than those in circulation through several mechanisms involving TGF-β1 and nitric oxide [40]. On the other hand, macrophages could be categorized as classically activated (M1) or alternatively activated (M2) macrophages. Whereas M1 macrophages express proinflammatory mediators, promote activation of cytotoxic CD8^+^ T cells, and foster antitumoral immune response, M2 macrophages induce angiogenesis, tissue remodeling, immune escape, and stimulate Treg cell differentiation by expression of immunosuppressive cytokines [41]. TAMs are recruited to the TME by different cytokines and often create a favorable milieu for tumor survival by suppressing immune responses. As in other malignancies, TAMs in HNSCC promote metastasis by inducing stemness and epithelial-mesenchymal transition (EMT) [42]. Indeed, two different meta-analyses showed that an infiltrate composed of a high number of CD163^+^ TAMs (M2-like) is associated with worse OS and PFS accompanied by poor cell differentiation and advanced disease status [43,44]. The aberrant activation of the STAT3 signaling pathway in HNSCC is associated with the infiltration of M2-like TAMs and MDSCs in the TME [45].

### 2.2. Development of a Tumor-Promoting Microenvironment

The TME comprises several cell types and a network of soluble factors that orchestrate the immunosuppressive microenvironment. Particularly, the TME in HNSCC is enriched in immune inhibitory molecules that can directly thwart host immunity, impacting survival or altering the function of immune cells [46,47,48]. Tumor-infiltrating Tregs are the primary source of both TGF-β1 and IL-10 [32,33]. While TGF-β1 significantly contributes to immune dysfunction in HNSCC, attenuating the activity of Teff cells and promoting the expansion of infiltrating Tregs (reviewed by Reference [49]), IL-10 is strongly associated with disease progression [50]. This cytokine exerts anti-inflammatory properties by favoring the recruitment and activity of other immunosuppressive cells.

The pleiotropic proinflammatory cytokine IL-6 can independently predict tumor recurrence, poor survival, and tumor metastasis in HNSCC [51]. Moreover, a recent study showed the association between IL-6 levels, survival, and immunosuppression. Notably, Tsai and colleagues observed that IL-6 upregulates PD-L1 expression and MDSCs infiltration in HNSCC tumors [52]. 

Bergmann and colleagues showed that prostaglandin-E2 (PGE2), a soluble product of cyclooxygenase-2 (COX-2) activity, is implicated in tumor immunosuppression in HNSCC, promoting the expansion of Tregs [53]. Interestingly, COX-2 expression was increased in HNSCC tumor biopsies compared to paired-normal tissue [54]. Therapeutic attempts to block PGE2-mediated immunosuppression in vivo using COX-2 inhibitors in a mouse model of HNSCC induced antitumor effects and restored antitumor immunity [55], thus highlighting its relevant role in immune escape. Additionally, COX-2 expression was higher in metastatic lesions and significantly correlated with tumor vascularization [56]. VEGF, a regulatory cytokine implicated in angiogenesis and vascularization, is also overexpressed in HNSCC tumors and is associated with poor prognosis [57].

### 2.3. Galectin-1 as a Soluble Immune Checkpoint

Galectin 1 (Gal-1) has been proposed as a novel immune checkpoint involved in different cancer hallmarks. In human tissue of HNSCC, Gal-1 overexpression has been associated with lower infiltration of T cells and was identified as an independent prognostic factor for shorter OS [58]. Interestingly, the presence of Gal-1 in tumor-associated exosomes secreted by head and neck cancer cell lines induced a marked suppression of CD8^+^ T-cell activity. This suppressive effect was characterized by the loss of CD27/CD28 co-stimulatory molecules and the decrease of IFN-γ production, in accordance with the phenotype presented by T cells on HNSCC tumor biopsies [59]. Further studies in a mouse model of HNSCC, showed that this lectin generates an immunosuppressive barrier that prevents T cell migration to the tumor, particularly by up-regulation of immune-inhibitory ligands including PD-L1 and Galectin-9 (Gal-9). In this regard, Gal-1 in this preclinical model improved the effect of anti-PD-1 immunotherapy. Accordingly, lower levels of Gal-1 in the tumor and stroma were associated with better response to anti-PD-1 therapy and higher survival rates in HNSCC patients with R/M disease [60]. Based on this evidence, Gal-1 has been proposed as a prognostic biomarker and a promising therapeutic target in HNSCC (Figure 1).

### 2.4. Alteration in the Antigen Presentation Machinery (APM)

One of the requirements for antitumor immunity is T cell recognition of tumor antigen peptides presented by the major histocompatibility complex (MHC), also known in humans as human leukocyte antigen (HLA), on the surface of tumor cells or antigen-presenting cells (APCs), a process known as antigen presentation. There are two classes of MHC molecules, the MHC class I proteins, which take part in antigen presentation to CD8^+^ T cells, and the MHC class II proteins, involved in antigen presentation to CD4^+^ T cells. Although immunogenic tumor antigens occur in HNSCC, in some cases they may not be recognized by T cells and trigger immune responses. In general, cells with complete loss of HLA evade T cell response but can trigger an immune response by natural killer (NK) cells. However, tumor cells may use different mechanisms to downregulate HLA expression to avoid recognition by NK or T cells. In this regard, the genomic and transcriptomic sequencing of HNSCC tumors revealed multiple mutations in HLA alleles and APM components [61]. Moreover, the downregulation of HLA class I antigen and LMP2 (a component of the APM) correlates with low CD8^+^ T cell infiltration and is significantly associated with lower survival rates in HNSCC patients [62], similar to other AMP components which are associated with poor clinical outcome [63]. 

Research efforts are currently focused on understanding the mechanisms implicated in these alterations. Recent findings have shown that EZH2, a histone-lysine N-methyltransferase enzyme, is inversely correlated with MHC-I expression in HNSCC. In vitro and in vivo blockade of EZH2 with small molecule inhibitors enhanced antigen presentation and antitumor immune responses. Based on these promising results, the combination of EZH2 inhibitor and anti-PD-1 mAb was proposed as an alternative therapeutic strategy to improve immunotherapy success in HNSCC [64].

## 3. Immune Checkpoint Pathways Implicated in HNSCC

Immune checkpoints are negative regulators of T cell activation while tuning the immune response to avoid its hyperactivation. The relevance of immune checkpoint pathways in tumor immune escape led to the design and implementation of several immunotherapeutic strategies to treat cancer. The major immune checkpoint pathways implicated in HNSCC immunosuppression are PD-1/PD-L1, cytotoxic T lymphocyte antigen 4 (CTLA-4), the T-cell immunoglobulin mucin 3 (TIM-3), the lymphocyte activation gene 3 (LAG-3), and T-cell immunoglobulin and immunoreceptor tyrosine-based inhibitory motif (TIGIT) (Figure 1).

### 3.1. CTLA-4

CTLA-4 is a cell surface molecule mainly expressed on T cells, especially Tregs and exhausted Teff cells. CTLA-4 interacts with CD80 (B7-1), and CD86 (B7-2) expressed on the surface of APCs. CTLA-4 binding to CD80 or CD86 inhibits T cell activation by inhibiting signaling through the TCR. The primary cells affected by the activation of this immune checkpoint are activated T cells in lymph nodes (LN), dendritic cells (DCs), and exhausted Teff cells. This molecule acts mainly through the downmodulation of CD4^+^ T helper function while enhancing Tregs activity [65]. CTLA-4 was the first negative regulator of T cell activation identified in the context of antitumor immunity, and its blockade using mAbs triggers tumor regression and a durable antitumor immunity in preclinical models [66].

CTLA-4 expression is increased in HNSCC tumor samples compared to normal tissue and is not correlated with LN metastasis or pathological tumor grade [67]. Interestingly, Yu and colleagues observed that CD8/CTLA-4 ratio negatively correlates with prognosis and the frequency of MDSCs and M2 macrophages regardless of HPV infection [67]. Likewise, previous studies have shown that both tumor-infiltrating and circulating Tregs upregulate CTLA-4 in HNSCC patients [32,35], being higher the frequency of CTLA-4^+^ cells on intra-tumoral Tregs as well as their immunosuppressive activity [31] through several mechanisms involving TGF-β1. Besides these observations in human samples, the blockade of CTLA-4 in an immune-competent mouse model of HNSCC significantly reduced MDSCs, M2 macrophages, and tumor burden while improving the effector function of T cells [67]. Similarly, the treatment with anti-CTLA-4 mAb in a preclinical model of HNSCC that recapitulates the tobacco-related molecular profile showed complete response and no tumor recurrence accompanied by an increase of CD8^+^ T-cell infiltration and a reduction in Tregs cell proportion [68]. Therefore, CTLA-4 expression by HNSCC tumors is an important immunosuppressive mechanism both in humans and mice and an attractive therapeutic target. In this regard, CTLA-4 blockade could improve clinical benefits of cetuximab based on the observation that CTLA-4^+^ Tregs increase with cetuximab treatment and via TGF-β-dependent mechanisms suppress cetuximab-mediated antibody-dependent cellular cytotoxicity, correlating with poor survival [69].

### 3.2. PD-1/PD-L1

PD-1 belongs to the CD28 family of co-stimulatory molecules expressed on the cell surface of tumor-infiltrating lymphocytes (TILs) such as Tregs, NK, T, and B cells. This co-inhibitory molecule interacts with the glycoprotein ligands PD-L1 and PD-L2. Both APCs and tumor cells express high amounts of PD-L1, which contribute to T-cell inhibition. The primary mechanism by which PD-1/PD-L1 interaction blunts immune responses involves inhibition of TCR intracellular signaling in Teff and Tregs cells, influencing T-cell survival, proliferation, and cytotoxicity. Chronic exposure of T cells to tumor antigens can lead to the expression of high levels of PD-1 on their surface, favoring a T-cell exhaustion profile [65]. Recently, Gao and colleagues have shown that nuclear PD-L1 regulates gene expression of several immune-related genes through its binding to the DNA. Mechanistically, PD-L1 translocates into the nucleus in an acetylation-dependent manner of the C-tail and binds directly to the DNA or interacts with specific transcription factors stimulating the expression of genes involved in inflammatory and antigen presentation pathways as well as enhancing the expression of other immune checkpoint inhibitors that could lead to acquired resistance to anti-PD-1/PD-L1 treatment [70]. Besides the relevant role of the PD-1/PD-L1 axis in mediated immunosuppression in the TME, Dammeijer et al. showed that this pathway could also be activated in tumor-draining lymph node (TDLN), which are enriched in tumor-specific PD-1 T cells abolishing the current idea that only occurs at the tumor site. In particular, PD-1/PD-L1 association in TDLN impair systemic antitumor response, and in melanoma patients, its abundance is associated with disease recurrence [71].

PD-L1 is expressed in histologically aggressive and T-cell enriched HNSCC tumors [72]. Moreover, PD-L1 expression has been associated with EMT in HNSCC patients [73], and although around 60% of HNSCC tumors express PD-L1, it is more frequent in HPV^+^ specimens [74]. Additionally, a higher infiltration of PD-1^+^ cells, mainly CD4^+^ and CD8^+^ T cells, in HPV^+^ tumors correlates with better survival [75]. Tumor-infiltrating and peripheral Tregs of HNSCC patients express high levels of PD-1 on the cell surface, triggering immunosuppression [35]. Interestingly, circulating PD-L1^+^-tumor-derived exosomes (TEX) are associated with disease progression (disease activity, tumor stage, and nodal involvement) in HNSCC patients. Moreover, PD-L1^+^-TEX can induce T cell dysfunction on activated CD8^+^ T cells in ex vivo assays and the in vitro blockade of these exosomes with an anti-PD-1 mAbs enhances antitumor immune response by reversing this immunosuppressive effects on CD8^+^ T cells [76]. 

Regarding the molecular mechanisms associated with PD-L1 expression in HNSCC, a recent study showed that PD-L1 is induced by cell-intrinsic and -extrinsic pathways downstream of EGFR and IFN-γ, both dependent on JAK2/STAT1. Indeed, in vitro studies showed that JAK2 inhibition impaired IFNγ-mediated PD-L1 upregulation at mRNA and protein level in tumor cells [74]. 

### 3.3. LAG-3

LAG-3 is a transmembrane CD4-related protein expressed on B cells, DCs, NK, and activated T cells surface. Particularly, Tregs express high amounts of LAG-3, which amplifies their immunosuppressive activity by enhancing IL-10 production. Like PD-1, LAG-3 is upregulated in exhausted T cells. Ligands for LAG-3 include MHC-II, Galectin-3 (Gal-3), and liver sinusoidal endothelial cell lectin (LSECtin) expressed by certain tumor types [65]. Gal-3 is expressed by both tumor and immune cells, mostly M2 macrophages [77]. LAG-3 inhibits CD4^+^ and CD8^+^ T cell effector functions and proliferation. It is thought that LAG-3 functions in coordination with other checkpoint molecules to promote T-cell dysfunction. However, the molecular mechanisms and pathways implicated in LAG-3 signaling are still under scrutiny. In this regard, the conserved KIEELE motif of the cytoplasmatic domain was shown to be indispensable for LAG-3 downstream signaling and inhibition of CD4^+^ T cells activation [78], and further studies demonstrated that MHC-II/LAG-3 triggers the activation of ITAM signaling in DCs, fostering a tolerogenic profile [79]. Therefore, MHC-II/LAG-3 interaction functions as a bidirectional inhibitory pathway. On the other hand, Gal-3/LAG-3 interactions modulate the innate branch of the antitumor immune response, mainly DCs and macrophages, and suppress CD8 ^+^T cell effector functions [80]. 

The role of LAG-3 in HNSCC has been studied for a long time in both preclinical or in vitro models and human samples. In a genetic mouse model of HNSCC characterized by complete loss of *Pten* and *Tgfb1r*, LAG-3 expression is increased on CD4^+^ and CD8^+^ T cells, as well as in Tregs. Furthermore, the in vivo administration of anti-LAG-3 mAb in this immunocompetent mouse model suppressed tumor growth and unleashed the antitumor CD8^+^ T cell-mediated response [81], supporting the relevant role of LAG-3 as an immune checkpoint and a possible therapeutic target in HNSCC. Likewise, LAG-3 is overexpressed on TILs of HNSCC patients, and its expression correlates with high pathological grade, tumor size, and worse prognosis [81]. It also correlates with CD8^+^ T cells and immunosuppressive cells, including Tregs, M2 macrophages, and MDSCs. A recent study using single-cell RNA sequencing (scRNA-Seq) from peripheral blood leukocytes of HNSCC patients showed that LAG-3 is expressed mainly on CD8^+^ T cells. A significant proportion of cancer patients express intracellular LAG-3 in peripheral CD8^+^ T cells, which correlates with the intracellular expression on matched TILs, poor prognosis, and decreased antitumor immune response [82].

### 3.4. TIM-3

TIM-3 is a negative co-stimulatory molecule expressed on CD4^+^ Th1, CD8^+^ T cells, Tregs, and other tumor cells. One of the TIM-3 ligands involved in immune regulation is Gal-9, which promotes apoptosis of Th1 cells, suppresses CD8^+^ T cell activity, and induces expansion of MDSCs [65].

In HNSCC, TIM-3 expression was observed in inflammatory cells, particularly CD8^+^ T cells and MDSCs, and is upregulated compared to normal mucosa and dysplasia. Furthermore, the high expression of TIM-3 is associated with LN metastasis and recurrence but not with OS. Expression of the TIM-3 ligand Gal-9 is not restricted to tumor cells, but it is also expressed on immune cells in the tumor stroma of HNSCC tumors. In this context, TIM-3/Gal-9 expression is closely associated with the Tregs and M2-like macrophages [83]. To further understand the role of TIM-3 in HNSCC development and progression, Liu and colleagues observed that the in vivo inhibition of TIM-3 in a preclinical immunocompetent mouse model controlled tumor growth by restoring the immune response, particularly increases the frequency of Teff in tumor and LN through modulation of TIM-3 expression in these cells and reduces the recruitment of MDSCs into the TME in a CXCL1 dependent-manner [84]. TIM-3 blockade also reduces Treg frequency and enhances IFN-γ production by CD8^+^ T cell [83].

Interestingly, in two orthotopic mouse models of HNSCC, the treatment with RT and anti-PD-1 mAb induced the upregulation of TIM-3 in CD8^+^ and CD4^+^ T cells, mainly in Tregs, mediating treatment resistance. Although targeting TIM-3 in addition to PD-L1 and RT results in tumor growth delay and improved survival through reducing intra-tumoral Tregs frequency, the response is not durable since remaining infiltrating Tregs are highly proliferative and could expand, fostering tumor recurrence [34]. Thus, TIM-3 expression is a mechanism of tumor evasion in HNSCC, and its blockade is an attractive immunotherapeutic target.

### 3.5. TIGIT

TIGIT is a co-inhibitory receptor expressed in lymphocytes, particularly in effector and regulatory CD4^+^ T cells, and in tumor cells. By binding to CD155, expressed on the surface of APCs, TIGIT dampens T cell hyperactivation, proliferation, and differentiation [65]. TIGIT is overexpressed on either peripheral T cells and tumor-infiltrating CD8^+^ and CD4^+^ T cells in HNSCC patients. Expression of the TIGIT ligand CD155 is higher in HNSCC than in normal tissue and is associated with worse OS and poor prognosis [85]. Thus, the blockade of this pathway may promote an active immunity against HNSCC tumors. Indeed, TIGIT inhibition significantly delayed tumor growth in HNSCC mouse models and enhanced antitumor immune responses by activating effector CD8^+^ T cells and reducing immunosuppressive cells, including Tregs and MDSCs [85]. High expression of TIGIT correlates with improved survival, only in HPV^+^ tumors [86].

## 4. Immune Checkpoint Blockade (ICB) in HNSCC Treatment

The use of mAbs targeting immune checkpoint pathways gave rise to modern immunotherapeutic modalities that revolutionized the treatment of cancer patients, including those with HNSCC. In 2016, the results from the KEYNOTE-012 trial guaranteed the approval of anti-PD-1 agent Pembrolizumab by the FDA for patients with R/M HNSCC with disease progression on or after platinum-containing chemotherapy regardless of HPV status. KEYNOTE-012 was an open-label, multicenter, phase Ib clinical trial testing Pembrolizumab in R/M HNSCC patients proving that it was well tolerated while inducing a strong antitumoral activity with an overall response rate of 18% [87,88]. These results were confirmed on the KEYNOTE-055 trial, which focused on R/M HNSCC after progression to platinum and cetuximab treatment [89]. Based on these results, two randomized phase III clinical trials evaluated the anti-PD-1 mAbs (Nivolumab or Pembrolizumab) in R/M HNSCC patients or HNSCC patients who progressed within six months to platinum-based chemotherapy, the CheckMate-141 and the KEYNOTE-040 trial, respectively. In the CheckMate-141 trial, Nivolumab was compared against the SoC systemic therapy (methotrexate, docetaxel, or cetuximab), showing longer OS on the Nivolumab arm with 7.5 months compared with 5.1 months with less toxicity [90], which allowed FDA to approve Nivolumab for R/M HNSCC patients. Likewise, the KEYNOTE-040 trial showed that Pembrolizumab-treated patients had increased OS compared to SoC, 8.4 months versus 6.9 months, respectively, and less treatment-related adverse events supporting its further testing as monotherapy or in combination in patients with early-stage disease. The median duration of response was 18.4 months in Pembrolizumab-treated patients compared with only 5.0 in the SoC arm, and the clinical benefit of Pembrolizumab (PFS and objective response rate (ORR)) relative to SoC was greater in those patients whose tumors express PD-L1 (Tumor proportion score (TPS) ≥ 50; or CPS ≥ 1) [91]. Later, in 2019, based on the KEYNOTE-048 results, Pembrolizumab was approved as monotherapy by the FDA as a first-line treatment for patients with PD-L1-positive R/M HNSCC or in combination with platinum and 5-fluorouracil in R/M HNSCC. In particular, KEYNOTE-048, a randomized, open-label phase III study, showed that Pembrolizumab alone or with chemotherapy improved median response duration by more than 16 and 2.5 months, respectively, versus cetuximab with chemotherapy. Moreover, Pembrolizumab monotherapy’s profound OS benefits were observed only in patients with PD-L1-positive tumors (CPS ≥ 1), while for Pembrolizumab with chemotherapy in all participants, regardless of the CPS score. Compared to standard therapy, the safety profile was favorable for Pembrolizumab alone or similar for Pembrolizumab with chemotherapy [92]. 

JAVELIN 100 is a randomized, double-blind, phase III clinical trial testing safety and antitumor efficacy of Avelumab, an anti-PD-L1 blocking monoclonal antibody, in combination with SoC chemoradiotherapy (SoC-CRT) against placebo plus SoC-CRT, as a first-line treatment in patients with locally advanced HNSCC being the primary endpoint PFS [93]. The study was completed recently, and the interim results presented in ESMO 2020 showed no significant improvement in PFS (based on 224 events) and OS (based on 131 events) with Avelumab plus SoC-CRT.

Besides the PD-1/PD-L1 axis, another checkpoint that has been widely explored as a therapeutic target is CTLA-4, with Ipilimumab and Tremelimumab being the two fully-humanized CTLA-4 antibodies most commonly used [94,95]. There is no anti-CTLA-4-based immunotherapy approved for the treatment of HNSCC. However, several ongoing clinical trials evaluate the combinations of anti-CTLA-4 antibodies with other immunotherapies or the current SoC HNSCC therapies. The hypothesis that the combination of therapies could yield better clinical benefits is being tested with good results in many cancer types. In this regard and based on previous observations of Durvalumab activity in R/M HNSCC, the EAGLE (NCT02369874) study is a phase III clinical trial testing the efficacy of Durvalumab, an anti-PD-L1 antibody alone or combined with Tremelimumab against SoC in patients with R/M HNSCC progressed after platinum-based therapy. The results recently published show no statistically significant differences in OS and PFS for Durvalumab or Durvalumab plus Tremelimumab versus SoC. Although ORR was similar among each arm, the only complete responses were observed in the immunotherapy ones. Furthermore, the median DoR was longer for Durvalumab and Durvalumab plus Tremelimumab compared to SoC, being 12.9, 7.4, and 3.7 months, respectively. Durvalumab showed antitumor activity, and the combination with Tremelimumab is tolerable despite not improving patient survival [96]. The NCT04080804 clinical trial is currently recruiting patients before surgery with locally advanced HNSCC to test if Nivolumab’s combination with Ipilimumab or Relatlimab (anti-LAG-3) potentiates antitumor immunity, thus enhancing the response to Nivolumab. Another therapeutic approach to block the LAG-3 immune checkpoint is by targeting its ligand Gal-3. In this regard, the NCT02575404 phase I clinical trial is currently testing the combination of Pembrolizumab with a specific Gal-3 inhibitor, GR-MD-02, in solid tumors, including HNSCC.

Several ongoing clinical trials are testing either the combination of different ICB or ICB with other therapies like chemoradiotherapy and vaccines. This is shortly discussed in the next section and summarized in Table 1. 

### 4.1. Combination of ICB with Other Therapies

Given the success of immunotherapy, several clinical trials are designed to test whether the efficacy of SoC CRT can be enhanced with the addition of different ICB. DURTRE-RAD trial (NCT03624231) is a two-arms, randomized, multicenter phase II study recruiting patients with non-resectable locally advanced HPV^-^HNSCC to test Durvalumab or Durvalumab and Tremelimumab in combination with radiotherapy as first-line treatment. As the combination of anti-PD-L1 and anti-CTLA-4 plus radiotherapy resulted in high toxicity, the trial is currently ongoing only for anti-PD-L1 alone or in combination with radiotherapy [97].

Based on the results of the KEYNOTE-040 and KEYNOTE-048, the KEYNOTE-412 (NCT03040999), a phase III clinical trial with 780 patients with locally advanced HNSCC, is taking place. Patients will be split in two groups; an experimental arm will be treated with a combination of Pembrolizumab, cisplatin, and radiotherapy; and the placebo arm will be treated with the same combination of cisplatin and radiotherapy, but with a placebo instead of Pembrolizumab. The primary hypothesis is that Pembrolizumab given in combination with SoC CRT is superior in event-free survival (disease progression or death in a maximum time frame of five years) than the SoC CRT. The investigators plan to complete the study by 2023 and believe that this trial may elucidate Pembrolizumab’s role when added to SoC in patients for whom five-year survival rates are poor, besides gaining insight on factors that influence the efficacy of immunotherapy by studying biomarkers data [98].

Besides the clinical trials previously discussed, there are others that are currently evaluating the combination of anti-PD-1 immunotherapies with targeted therapies such as Cetuximab (NCT03082534), Lenvatinib (a multikinase inhibitor of VEGFR 1–3, FGFR 1–4, PDGFRa, RET, and KIT; NCT02501096), Itacitinib (PI3K-delta inhibitor, NCT02646748), among others. Moreover, a clinical trial evaluating Acalabrutinib, a BTK inhibitor, has just been completed (NCT02454179).

### 4.2. Combination of ICB with Other Immunotherapies

Several clinical trials are testing the combination of PD-1/PD-L1 immune checkpoint inhibition with vaccination. The use of cancer vaccines as treatment is based on providing antigens to activate the immune system, promote an antitumor immune response, or overcome immunosuppression. The NCT02426892 phase II clinical trial is testing the combination of ISA101, a synthetic long-peptide HPV-16 vaccine inducing HPV-specific T cells, with Nivolumab. The study aims to determine whether this combination amplifies the anti-PD-1 treatment efficacy in patients with HPV^+^ HNSCC. To summarize, the overall response rate observed was 33% and a median OS of 17.5 months; these were promising results when compared to PD-1 monotherapy. However, the antitumor effects of the vaccine are yet to be studied [99]. Another clinical trial evaluating the safety and efficacy of an HPV-vaccine is the NCT03260023. Specifically, the study tests the combination of Avelumab with TG4001 in patients with R/M HPV-16-positive cancer. Preliminary results showed that the combination is safe and provides antitumor immunity. There is a shift in gene expression signatures towards a more active innate and adaptive immune response [100]. 

The use of oncolytic viruses is an immunotherapy strategy that utilizes a virus to replicate in tumor cells resulting in their lysis, leading to an immunogenic cell death and subsequent activation of the antitumor immune response. In this regard, there is an ongoing clinical trial, the MASTERKEY232/KEYNOTE-137 (NCT0262600), which aims to evaluate the safety and efficacy of Pembrolizumab in combination with TVEC, a genetically modified type I herpes simplex virus, in R/M HNSCC. The first results demonstrated the safety and clinical activity for the combination under study.

### 4.3. ICB in the Neoadjuvant Setting

Around 50% of patients with locally advanced HNSCC develop recurrence and/or distant metastasis after definitive local therapy. Although there have been many attempts to improve survival with induction or neoadjuvant chemotherapy, none have demonstrated efficacy. Due to the effectiveness of ICB in R/M disease, most of the ongoing clinical trials testing neoadjuvant settings have immunotherapy as a foundation of the study (reviewed by Reference [101]). 

NCT02296684 was one of the first clinical trials testing Pembrolizumab as neoadjuvant or adjuvant therapy in high-risk patients with locally advanced HNSCC treated with the current SoC surgical approaches. The results showed that Pembrolizumab was safe and there was low locoregional recurrences or distant metastasis at the time of the one-year follow-up [102]. Likewise, preliminary results of the phase I/II Checkmate-358 trial suggest that patients can benefit from Nivolumab in the neoadjuvant setting regardless of their HPV status [103]. Furthermore, the NCT02641093 phase II trial testing Pembrolizumab with cisplatin and radiation as adjuvant therapy in resected HNSCC shows promising results and suggests that pathological response is accompanied by robust immune cell infiltration at surgery time [104]. Besides these trials focused on PD-1 inhibitors, others are blocking PD-L1. In this regard, IMvoke010 (NCT03452137), an ongoing phase III clinical trial, aims to evaluate the efficacy in terms of PFS and the safety of Atezolizumab as adjuvant therapy in patients with high-risk locally advanced HNSCC [105].

Based on the synergism of the dual blockade of PD-1 and CTLA-4, many clinical trials are testing this combination. For instance, the phase III clinical trial IMSTAR-HN (NCT03700905) is currently recruiting patients with locally advanced HNSCC for the treatment with Nivolumab in combination with Ipilimumab in the adjuvant/neoadjuvant setting [106].

In addition to these studies, there are numerous other ongoing clinical trials. In summary, neoadjuvant immunotherapy has shown an acceptable safety profile, being well-tolerated in HNSCC patients with promising efficacy. However, many questions remain to be answered, such as adequate dosing and duration of treatment. 

## 5. Biomarkers of Response to Immunotherapy in HNSCC

Nivolumab and Pembrolizumab have been approved for treating patients with R/M disease, most HNSCC patients’ progress to therapy. Therefore, there is an urgent need to develop and validate robust predictive biomarkers, to improve the selection of patients who will receive clinical benefit [107]. Indeed, integrating biomarkers of response to immunotherapy is proposed to correctly select the best treatment options for each patient [108]. In the following sections, we discuss biomarkers that have been evaluated in HNSCC. 

### 5.1. Tumor Genomic Features: Microsatellite Instability and Tumor Mutational Burden

Microsatellite Instability (MSI) is an underlying genetic process contributing to high Tumor Mutation Burden (TMB) and consequent neo-antigen formation. TMB and MSI are indirect measures of tumor antigenicity generated by somatic mutations. Therefore, they are both predictive biomarkers of response to immunotherapy treatment in several malignancies. 

Tumors with high MSI show improved response to checkpoint inhibition, an effect which was first demonstrated in patients with metastatic colorectal carcinoma treated with anti-PD-1 therapy [109]. Accumulating evidence suggests that both genomics of MSI-positive tumors and their respective microenvironment enriched in CD8 T cells contribute to high response rates to immunotherapy [110]. In 2017, the FDA approved Nivolumab and Pembrolizumab to treat MSI-positive cancers of any histology [111]. Despite the low frequency of MSI in HNSCC tumors, the Society of Immunotherapy of Cancer (SITC) recommends its testing to predict response to PD-L1 inhibitors [107]. Several studies and meta-analyses propose the TMB as a promising biomarker in HNSCC and other solid malignancies like non-small-cell lung cancer (NSCLC) and melanoma, where there is a positive correlation between OS and response rates to immunotherapy [112,113]. Despite these observations, the SITC has not currently recommended TMB analyses for HNSCC [107]. However, the FDA approved in June 2020 the use of Pembrolizumab for the treatment of previously treated solid tumors including anal, biliary, cervical, endometrial, mesothelioma, neuroendocrine, salivary, small-cell lung, thyroid, and vulvar cancers patients with high TMB (TMB-H) [≥10 Mutations/Megabase (Mut/Mb)].

### 5.2. Tobacco Smoke and HPV Status

Tobacco smoking is one of the major risk factors for the development and progression of HNSCC and is an important prognostic factor for survival and mortality after diagnosis [6]. Tobacco smoking generates DNA damage, induces mutations, and modifies the tumor immune microenvironment. Particularly, HNSCC tumors that harbor a genetic smoking signature have a lower immune infiltrate, local immunosuppression, and diminished IFN-γ signaling and cytolytic activity [22,114,115]. Furthermore, a smoking signature correlates with poor OS and high TMB load in HNSCC tumors [22,115]. Even though the smoking signature is positively associated with response to ICB in NSCLC [116], there is little evidence in the literature correlating smoking status and response to immunotherapy in HNSCC. Interestingly, for two cohorts of HNSCC patients treated with anti-PD-1/PD-L1 inhibitors, never-smokers have higher clinical benefit than current/former smokers [90,115].

Besides tobacco smoking, HPV infection is another significant risk factor for the development of HNSCC. Indeed, the HPV infection is related to around 20–25% of all HNSCC cases, and they are clinically and biologically distinct from their non-viral counterparts [117]. HPV^+^ HNSCC tumors, particularly OSCC, are associated with good prognosis [7]. Interestingly, smoker status is associated with worse OS in HPV^+^ tumors. This may be explained by the genetic alterations induced by tobacco-associated carcinogens rendering HPV^+^ tumors less responsive to therapy [7]. The HPV infection in HNSCC promotes an inflamed immune microenvironment with higher CD8 T cell activation, Tregs, and greater levels of CTLA-4 expression [22]. However, there is no robust evidence supporting differential clinical benefit despite the effects of HPV status in the TME. Therefore, the SITC does not recommend using the HPV status to stratify HNSCC patients [107].

### 5.3. PD-L1 Expression

PD-L1 expression on the tumor and immune cells has been associated with improved treatment outcomes to anti-PD-1/PD-L1 immunotherapy. Indeed, it has been approved by the FDA as a predictive biomarker of response in several cancers such as NSCLC, bladder, and cervical cancer, among others [118]. Specifically, in HNSCC, PD-L1 expression by IHC on tumor and infiltrating immune cells has been widely studied as a biomarker of response to PD-1/PD-L1 ICB. Furthermore, current evidence supports that the CPS offers a higher predictive value than the TPS [107]. In this regard, patients enrolled in the KEYNOTE-012 showed no differences in ORR between PD-L1^+^ and PD-L1^-^ patients when using a TPS ≥ 1. However, with CPS, PD-L1^+^ patients had increased ORR. The relevance of CPS as a biomarker was also demonstrated in the KEYNOTE-048 where patients with CPS ≥ 20 and CPS ≥ 1 benefit from Pembrolizumab monotherapy while for Pembrolizumab plus chemotherapy was independent of the PD-L1 status. Likewise, the KEYNOTE-040 demonstrated that patients treated with Pembrolizumab with a CPS ≥ 1 have an improved OS. However, some patients negative for PD-L1 expression still benefit from therapy, suggesting that the predictive value of PD-L1 expression is not absolute. In this regard, a challenge to use PD-L1 as a biomarker in HNSCC is the intra- and inter-tumor heterogeneity [107]. Rasmussen and colleagues observed in 16 HNSCC patients that PD-L1 expression varies within the tumor, affecting both the TPS and CPS [119]. Similarly, in NSCLC, PD-L1 expression has been recently associated with the anatomical site of the biopsy, and its predictive value is also different for each site [120]. On the other hand, a recent comparative study among three different PD-L1 IHC assays showed moderate concordance among the assays and considerable differences in PD-L1 positivity [121]. The recent findings suggest that TEXs carrying functional PD-L1 can induce immunosuppression inhibiting Teff cells [122]. In this regard, Theodoraki and colleagues observed that high levels of TEX- PD-L1^+^ at baseline and its reduction during treatment are associated with response in HNSCC patients [123]. Thus, they have been recently proposed as potential non-invasive biomarkers to monitor patients’ response to treatment [122,123].

### 5.4. T-Cell Inflamed Gene Expression and Novel Insights into the TME

A T-cell inflamed gene expression signature has been significantly associated with overall response and PFS to anti-PD-1 therapy in HNSCC and in other solid tumors like melanoma. The T-cell inflamed signature comprises IFN-γ-responsive genes related to antigen presentation, chemokine expression, cytotoxic activity, and adaptive immune resistance. Thus, this T-cell inflamed profile could be a useful biomarker based on its high predictive value [124]. 

Apart from T cells, other tumor-infiltrating immune cells like macrophages, myeloid cells, and NK cells are also relevant in modulating the TME improving tumor responses to treatment [25]. The use of bioinformatic tools to infer the composition of the immune infiltrate from transcriptomic data of tumor biopsies is becoming more relevant and useful. Moreover, scRNA-seq studies are critical to further understand the underlying molecular mechanisms of response and resistance to treatment at a cellular level. By applying bioinformatic methods on tumor biopsies sequencing data, researchers hope to improve biomarkers’ precision to stratify patients better and propose novel therapeutic targets. A recent study by Chen and colleagues shows a novel immune molecular classification of HNSCC with implications on immunotherapy response. Based on immune-related gene signatures representing immune status or immune cells, the authors [125] described two subgroups on The Cancer Genome Atlas (TCGA) HNSCC cohort and other six independent cohorts. While the Active Immune class was enriched in B-cell and M1 macrophages infiltration and cytolytic activity (markers of immune activation), the Exhausted Immune class was enriched in activated-stroma, M2 macrophages infiltration, and TGF-β1 expression (markers of immune exhaustion). When these classes were studied on a melanoma cohort of patients treated with anti-PD-1, the Active Immune class was associated with patients that responded to therapy showing its good predictive value [125]. 

The development of single-cell genomics methods in the last years has provided new ways to explore the TME. Specifically, scRNA-Seq experiments reveal the inter- and intra-tumoral heterogeneity in cancer cells and their association with different stromal and immune cells in the TME [126]. In a scRNA-Seq study of primary tumors and LN metastasis biopsies from HNSCC patients, Puram and colleagues [127] found consistent immune cell populations and expression profiles across patients, while they differ their proportions in the TME. In the T cell compartment, they found four sub-populations (Tregs, conventional CD4^+^ T helper cells, and two cytotoxic CD8^+^ T cells) differing in the expression of co-inhibitory molecules. Interestingly, the gene expression profiles of cytotoxic CD8^+^ T cells revealed an exhaustion program characterized by the expression of different markers of T cell dysfunction and exhaustion such as LAG-3, TIGIT, PD-1, and CTLA-4. Notably, the proportion of CD8^+^ T cells varied between patients. This suggests a potential association with response to ICB and biomarkers in HNSCC. Furthermore, the study of malignant cells revealed that they clustered accordingly to their tumor of origin and presented differential expression of common signatures within each tumor, reflecting inter- and intra-tumoral heterogeneity [127].

Additional scRNA-Seq studies in tumor biopsies are necessary to reveal associations between the TME heterogeneity and resistance to treatment in HSNCC. The study of the immune cell compartment and its crosstalk with tumor cells and the TME is crucial to establish new biomarkers of response to immunotherapy and understanding the mechanisms underlying resistance at a cellular level.

### 5.5. Microbiota as a Potential Novel Biomarker of Response to Immunotherapy

The oral microbiome is affected by common HNSCC risk factors, including HPV infection and tobacco smoking [128]. A retrospective study associated the oral abundance of certain bacteria with reduced HNSCC development risk [129]. For several cancer malignancies, the composition of the microbiota has been associated with immune dysregulation, cancer progression, and response to treatment [130,131]. Indeed, several shreds of evidence support the role of the gut microbiome in regulating the antitumor immune response and the success of immunotherapy in melanoma and hepatocellular carcinoma, among others [132,133]. The role of the microbiota in HNSCC and response to immunotherapy remains to be thoroughly studied. Only the CHECKMATE-141 clinical trial explored the oral microbiota composition in patients treated with Nivolumab, but no significant correlation with response to treatment or survival was observed.

### 5.6. Novel Insights in Biomarkers: Combined and Integrative Strategies

The current understanding of the clinical response to ICB suggests that any single biomarker cannot effectively identify patients due to the complexity of the tumor biology and immune response. Therefore, combining predictive biomarkers holds promise to be more effective and specific as a strategy to infer, identify and capture the immune status of tumors [108]. Cristescu et al. showed for different cancer types, including HNSCC, that TMB, PD-L1 expression, and T-cell inflamed gene expression profiles measure different aspects of the tumor immune response, as they are independent markers to predict the efficacy of anti-PD-1/PD-L1 therapy [113]. This suggests that every single biomarker provides complementary information of the TME, and its use in an integrated way will improve the identification of patients who will respond to immunotherapy. Indeed, ongoing efforts are currently focused on developing multifactorial predictive models by integrating data relative to different features of both tumor biology and tumor immune response. For instance, Wang and colleagues have recently developed a method to measure the tumor immunogenicity score (TIGS) that refers to the tumor antigenicity and its ability to present antigens to the immune cells. Particularly, the TIGS score combines TMB as a measure of tumor antigenicity and an antigen processing/presentation gene expression signature. Interestingly, the TIGS score has a better performance than TMB alone to predict response to ICB and correlates with ORR; and is also better than other biomarkers based only on gene expression profile (i.e., T-cell inflamed gene expression) [134]. Likewise, the immunophenoscore is a machine learning algorithm that generates an aggregated score from the expression of MHC molecules, immunomodulators, effectors, and suppressor cells with usefulness to predict survival and response [135]. Another successful strategy is multiplex IHC (mIHC) and multiplex immunofluorescence (mIF) for the study of the TME. In contrast to previously described biomarkers, mIF/mIHC allows the visualization of several markers in the same tumor section, thus providing spatial and co-expression information. A recent systemic review and meta-analysis involving more than 10 cancer types, including HNSCC, has shown that both mIHC/mIF had higher diagnostic accuracy in predicting clinical response to anti-PD-1/PD-L1 immunotherapy than PD-L1, TMB, or different gene expression signatures alone. Interestingly, its predictive value is similar to that reached by the combination of PD-L1 IHC, TMB, and/or gene expression [136]. 

In summary, the multidimensional examination of the TME by different approaches enhances the characterization and understanding of tumor heterogeneity thus enhancing the value in predicting immunotherapy treatment response. However, further studies and clinical trials are needed to confirm their biomarker potential.

## 6. Conclusions and Future Directions

HNSCC is one of the most highly immune-infiltrated cancer types. However, its immune microenvironment has immunosuppressive features and HNSCC tumors can successfully escape the antitumor immune response. Among the several mechanisms implicated in immune evasion, the expression of different immune checkpoint pathways has been of great interest due to the development of novel immunotherapeutic strategies targeting these pathways, which revolutionized the treatment of solid tumors, including HNSCC. 

The only ICB approach approved to treat HNSCC involves inhibition of the immune checkpoint pathway PD-1/PD-L1 by the administration of mAbs against PD-1. Numerous ongoing clinical trials are interrogating the potential clinical benefit of its combination with other types of immune modulators or with other agents. Importantly, there are still many unresolved challenges to improve patient outcomes and the number of patients who will benefit from these therapies. Consequently, there is an urgent need for predictive biomarkers of response and prognosis. In this regard, PD-L1 expression, T-cell inflamed profile, and the TMB have been proposed as promising predictive biomarkers for HNSCC, while other candidate biomarkers not discussed in this review, such as circulating tumor cells and circulating tumor DNA, are under investigation. Furthermore, ongoing efforts are focused on biomarker integration strategies to enhance the predictive power by considering several tumor features at the same time. A novel and promising immunotherapeutic approach use autologous chimeric antigen receptor (CAR) T cells. Indeed, there is an ongoing phase I clinical trial for patients with locally advanced or recurrent disease (NCT01818323) testing the efficacy of the intra-tumoral administration of genetically engineered T-cells to express the following: T1E28z, second-generation of CAR containing a promiscuous ligand of ErbB; and 4αβ, a chimeric IL-4 receptor. By this approach, researchers aim to block ErbB receptors, which are known key players in the initiation of several solid tumors, including HNSCC, and favor the ex vivo T-cell expansion by IL-4 [137]. Both in vitro and preclinical studies revealed the antitumor activity of these T-cells. Many investigations are still underway, to identify and validate tumor-associated antigens and how to overcome the challenges associated with CAR T cells in treating solid tumors [138]. In this regard, Mei and colleagues have recently proposed mucin-1 (MUC-1) as a target for CAR-T cells based on the higher expression of this protein by tumor cells, compared to non-neoplastic tissue and the effective cytotoxicity of MUC-1 CAR-T cells in vitro and in vivo [139]. 

Due to the high heterogeneity of this cancer type, it is difficult to expand one therapeutic option to all patients. Basic, translational, and clinical research will be required to improve the treatment and management of HNSCC patients.

## Figures and Tables

**Figure 1 cancers-13-01018-f001:**
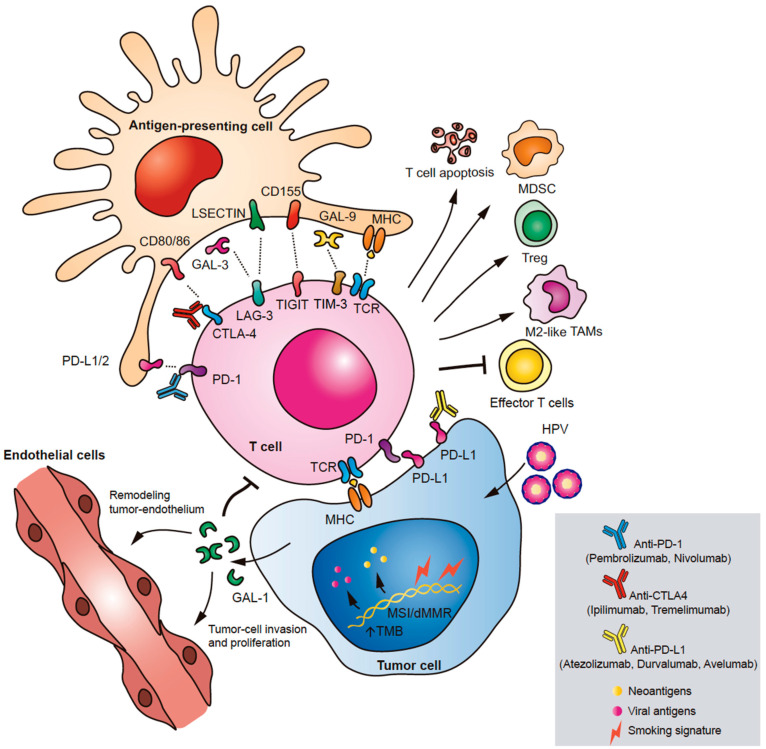
Immune checkpoints in head and neck squamous cell carcinoma (HNSCC). The main immune checkpoint pathways involved in HNSCC immune escape are illustrated: PD-1/PD-L1, CTLA-4, TIM-3, LAG-3, and TIGIT. Through different molecular mechanisms and signaling pathways, immune checkpoint molecules promote apoptosis of T cells, inhibit the effector function of T cells, and induce expansion of immunosuppressive MDSCs, M2-like TAMs, and Tregs. A novel soluble immune checkpoint implicated in HNSCC is Gal-1. Gal-1 is expressed by tumor cells triggering immune escape, modulation of tumor endothelium, and metastasis.

**Table 1 cancers-13-01018-t001:** Ongoing clinical trials testing Immune Checkpoint Blockade combinations. AEs, adverse events; DCR, disease control rate; DFS, disease-free survival; DoR, duration of response; DRR, durable response rate; ICI, immune checkpoint inhibitors; ORR, objective response rate; OS, overall survival; PBLs, Peripheral Blood Lymphocytes; PFS, progression-free survival; QOL, quality of life; SAEs, serious adverse events; SoC, standard of care; TIL, Tumor Infiltrating Lymphocyte.

Clinical Trial NCT Number	Clinical Trial Title	Status	Interventions	Immune Checkpoint Tested	Clinical Trial Details	Enrollment (Number of Patients)
NCT04454489	Quad Shot Radiotherapy in Combination with Immune Checkpoint Inhibition	Not yet recruiting	Quad-shot palliative radiotherapy + Pembrolizumab	PD-1	Phase: Phase II Intervention Model Description: Single group assignment, non-randomized Outcome Measures: ORR Response rate in the target lesions and in the non-target lesions DoR at the target lesions Percentage of participants with PFS and OS Incidences of AEs	15
NCT03313804	Priming Immunotherapy in Advanced Disease with Radiation	Recruiting	Nivolumab+ Radiation (Stereotactic Body Radiation Therapy OR fractionated radiation therapy) Pembrolizumab+ Radiation (Stereotactic Body Radiation Therapy OR fractionated radiation therapy) Atezolizumab + Radiation (Stereotactic Body Radiation Therapy OR fractionated radiation therapy)	PD-1/ PD-L1	Phase: Phase II Intervention Model Description: Single group assignment, non-randomized Outcome Measures: Six-month PFS Percentage of PD-1^+^ CD4^+^ T cells andPD-1^+^ CD8+ T cells prior to treatment versus with concurrent treatment Percentage of CD8^+^ T-cells that are gamma-interferon positive during treatment Percentage PD-L1^+^ CD4^+^ and PD-L1^+^CD8^+^ T-cell expression differences during treatment	57
NCT03228667	QUILT-3.055: A Study of Combination Immunotherapies in Patients Who Have Previously Received Treatment with Immune Checkpoint Inhibitors	Recruiting	Cohort 1: Patients who progressed on or after a single-agent ICI after experiencing an initial CR or PR. N-803 + Pembrolizumab or Nivolumab or Atezolizumab or Avelumab or DurvalumabCohort 2 and Cohort 3 only for selected NSCLC patients. Experimental Cohort 4: Patients who are currently receiving PD-1/PD-L1 checkpoint inhibitor therapy and have disease progression after experiencing SD for at least 6 months during their previous treatment with PD-1/PD-L1 checkpoint inhibitor therapy N-803 + Pembrolizumab or Nivolumab or Atezolizumab or Avelumab or Durvalumab Experimental Cohort 5: Patients who experienced disease progression by Investigator-assessment per irRECIST while receiving treatment in Cohorts 1-4 N-803 + Pembrolizumab or Nivolumab or Atezolizumab or Avelumab or Durvalumab + PD-L1 t-haNK	PD-1/ PD-L1	Phase: Phase II Intervention Model Description: Parallel assignment, non-randomized Outcome Measures: ORR Disease-specific survival OS Time to Response DoR Incidence of AEs QOL PFS	636
NCT03522584	Durvalumab, Tremelimumab and Hypofractionated Radiation Therapy in Treating Patients with Recurrent or Metastatic Head and Neck Squamous Cell Carcinoma	Recruiting	Durvalumab + Tremelimumab + Radiation (Hypo fractionated radiation therapy using either HIGRT or SBRT)	PD-L1 CTLA-4	Phase: Phase I Phase II Intervention Model Description: Single group assignment Outcome Measures: Incidence of AEs ORR PFS OS	20
NCT03544723	Safety and Efficacy of p53 Gene Therapy Combined with Immune Checkpoint Inhibitors in Solid Tumors.	Recruiting	Ad-p53 with anti-PD-1/anti-PD-L1 (based on FDA approved)	PD-1	Phase: Phase II Intervention Model Description Parallel assignment, non-randomized Outcome Measures: ORR Safety assessments of AES Preliminary assessment of DoR by RECIST 1.1 Preliminary assessment of PFS by RECIST 1.1	40
NCT04282109	Nivolumab in Combination with Paclitaxel in Subjects with Head and Neck Cancer Unable for Cisplatin- Study to Assess the Efficacy and Safety of based Chemotherapy (NIVOTAX)	Recruiting	Nivolumab + Paclitaxel Cetuximab + Paclitaxel (Active comparator)	PD-1	Phase: Phase II Intervention Model: Parallel assignment, randomized Outcome Measures: PFS ORR DCR DoR Rate of progressive disease (PD) at 6 months Two-year OS Five-year OS Others	141
NCT03944915	De-Escalation Therapy for Human Papillomavirus Negative Disease	Recruiting	Carboplatin + Paclitaxel +Nivolumab (Induction Therapy) Radiation therapy with Chemotherapy	PD-1	Phase: Phase II Intervention Model: Single group assignment, non-randomized Outcome Measured: DRR PFS OS Locoregional and distant control after completing chemoradiation	36
NCT03426657	Radiotherapy with Double Checkpoint Blockade of Locally Advanced HNSCC	Recruiting	Durvalumab + Tremelimumab + Radiation therapy	PD-L1 CTLA-4	Phase: Phase II Intervention Model: Prospective, open-label, non-randomized Outcome Measured: Number of participants receiving the protocol treatment until cycle 6 of antibody treatment Predictive character of changes of CD8^+^ tumor infiltrating immune cells after induction chemoimmunotherapy Absence of any dose-limiting toxicities PFS	120
NCT03841110	FT500 as Monotherapy and in Combination with Immune Checkpoint Inhibitors in Subjects with Advanced Solid Tumors	Recruiting	FT500 Monotherapy FT500 in Combination with Nivolumab, Pembrolizumab or Atezolizumab	PD-1/ PD-L1	Phase: Phase I Intervention Model: Parallel assignment, non-randomized Outcome Measured: The incidence of subjects with dose-limiting toxicities within each dose level cohort to determine the Maximum Tolerated Dose ORR Duration of FT500 persistence	76
NCT03377400	Definitive CCRT Combined with Durvalumab and Tremelimumab for Inoperable Esophageal Cancer	Active, not Recruiting	5FU/CDDP + Durvalumab/Tremelimumab	PD-L1 CTLA-4	Phase: Phase II Intervention Model: Single group assignment. Outcome Measured: PFS	40
NCT03673735	Maintenance Immune Check-point Inhibitor Following Post-Operative Chemo-radiation in Subjects with HPV-negative HNSCC	Not yet recruiting	Durvalumab before Chemoradiotherapy and for 6 months every 4 weeks after CRT Control: placebo before CRT and for 6 months every 4 weeks after CRT Radiotherapy	PD-L1	Phase: Phase III Intervention Model: Parallel assignment, randomized Outcome Measured: DFS OS Cumulative incidence of distant metastases, locoregional recurrence and second cancers (all sites) Rate of toxicity assessed by clinicians QOL	650
NCT04393506	Inductive Camrelizumab and Apatinib for Patients with Locally Advanced and Resectable Oral Squamous Cell Carcinoma	Recruiting	Camrelizumab and Apatinib, followed by radical surgery and post-operative radiotherapy/chemoradiotherapy	PD-1	Phase: Phase I Intervention Model: Sequential assignment. Outcome Measured: Major pathologic response Two-year OS Two-year tumor recurrence rate	20
NCT03946358	Combination of UCPVax Vaccine and Atezolizumab for the Treatment of Human Papillomavirus Positive Cancers (VolATIL)	Recruiting	Atezolizumab + UCPVax	PD-L1	Phase: Phase II Intervention Model: Single group assignment Outcome Measures: ORR at 4 months OS PFS Health-related quality of life (HrQoL)	47
NCT04058145	AMD3100 Plus Pembrolizumab in Immune Checkpoint Blockade Refractory Head and Neck Squamous Cell Carcinoma	Recruiting	Pembrolizumab + AMD3100 Q3w Pembrolizumab + AMD3100 weekly Pembrolizumab + AMD3100	PD-1	Phase: Phase II Intervention Model: Parallel assignment, randomized. Outcome Measures: Participants that experience a dose limiting toxicity ORR PFS OS Participants with AEs Duration of response	57
NCT04080804	Study of Safety and Tolerability of Nivolumab Treatment Alone or in Combination with Relatlimab or Ipilimumab in Head and Neck Cancer	Recruiting	Nivolumab + Relatlimab Nivolumab + Ipilimumab Nivolumab	PD-1 CTLA-4 LAG-3	Phase: Phase II Intervention Model: Parallel assignment, randomized. Outcome Measures: AEs related to monotherapy or combinations Radiographic response Levels of TIL subset Levels of PBLs Effector CD4^+^ and CD8^+^ T cells TMB Gene expression signature	60
NCT03690986	VX15/2503 in Combination with Ipilimumab or Nivolumab in Patients with Head and Neck Cancer	Recruiting	Group A: VX15/2503 Group B: VX15/2503 + Ipilimumab) Group C: VX15/2503 + Nivolumab) Group D: Nivolumab Group E: Ipilimumab Group F: no treatment	PD-1 CTLA-4	Phase: Phase I Intervention Model: Parallel assignment, randomized Outcome Measures: Change in immune profile in the TME Change in circulating percentage of immune suppressor subsets in peripheral blood Phenotypic shifts in T-lymphocyte subsets in peripheral blood Incidence of AEs	36
NCT02718820	Pembrolizumab Plus Docetaxel for the Treatment of Recurrent or Metastatic Head and Neck Cancer	Active, not recruiting	Docetaxel + Pembrolizumab	PD-1	Phase: Phase I Phase II Intervention Model: Single group assignment Outcome Measures: Overall response rate Best overall response rate Individual duration of response over time QoL OS PFS AEs	22
NCT03684785	Intratumoral Cavrotolimod Combined with Pembrolizumab or Cemiplimab in Patients with Advanced Solid Tumors	Recruiting	Dose escalation phase 1b: AST-008 + Pembrolizumab Dose expansion phase 2, Meker cell carcinoma, patients that have progressed on an anti-PD-1/PD-L1 therapy or are otherwise refractory to CPI therapy: AST-008 + Pembrolizumab Dose expansion phase 2, cutaneous squamous cell carcinoma, patients that have progressed on an anti-PD-1/PD-L1 therapy or are otherwise refractory to CPI therapy: AST-008 + Cemiplimab	PD-1	Phase: Phase Ib Phase II Intervention Model: Sequential Assignment, randomized Outcome Measures: AEs Disease assessment wit RECIST 1.1 ORR Safety evaluation of AST-008 alone or in combination PFS and OS Pharmacokinetic parameters Disease control rate Others	130
NCT03212469	A Trial of Durvalumab and Tremelimumab in Combination with SBRT in Patients with Metastatic Cancer (ABBIMUNE)	Recruiting	Durvalumab + Tremelimumab + SBRT radiation	PD-1 CTLA-4	Phase: Phase I Phase II Intervention Model: Parallel assignment, non-randomized Outcome Measures: Dose-limiting toxicity	55
NCT03818061	Atezolizumab and Bevacizumab in Patients with Recurrent or Metastatic Squamous-Cell Carcinoma of the Head and Neck (ATHENA)	Recruiting	Atezolizumab + Bevacizumab in HPV^+^ or HPV^-^	PD-L1	Phase: Phase II Intervention Model: Parallel assignment, non-randomized Outcome Measures: ORR Disease control rate Best OR DoR PFS OS Others: immune cell characterization, microbiome analysis, immuno-phenotyping	110
NCT03829501	Safety and Efficacy of KY1044 and Atezolizumab in Advanced Cancer	Recruiting	KY1044 monotherapy phase 1 (dose escalation) KY1044 + Atezolizumab phase 1 (dose escalation) KY1044 monotherapy phase 2 KY1044 + Atezolizumab phase 2	PD-L1 ICOS	Phase: Phase I Phase II Intervention Model: Sequential assignment, non-randomized Outcome Measures: AEs and SAEs Number of dose interruptions, reductions and dose intensity ORR Survival rate PFS DoR OS Others: immune cell characterization, microbiome analysis, immuno-phenotyping	412
NCT02551159	Phase III Open Label Study of MEDI 4736 with/without Tremelimumab Versus Standard of Care (SoC) in Recurrent/Metastatic Head and Neck Cancer	Active, not recruiting	MEDI4736 monotherapy MEDI4736+Tremelimumab SoC	PD-L1 CTLA-4	Phase: Phase III Intervention Model: Parallel assignment, randomized Outcome Measures: Efficacy of MEDI4736 monotherapy compared to SoC in terms of OS The efficacy of MEDI4736 + Tremelimumab combination therapy compared to SoC in terms of OS, ORR, PFS, Second Progression (PFS2) and DoR The efficacy of MEDI4736 + Tremelimumab combination therapy compared to SoC in terms of best objective response (BoR), Time to First Subsequent Therapy (TFST), and Time to Second Subsequent Therapy (TSST) Others: immune cell characterization, microbiome analysis, immuno-phenotyping	823
NCT03517488	A Study of XmAb®20717 in Subjects with Selected Advanced Solid Tumors (DUET-2)	Recruiting	XmAb20717 (bispecific antibody)	PD-1 CTLA-4	Phase: Phase I Intervention Model: Sequential assignment, randomized Outcome Measures: Determine the safety and tolerability profile of XmAb20717	154
NCT03693612	GSK3359609 Plus Tremelimumab for the Treatment of Advanced Solid Tumors	Recruiting	GSK3359609+ Tremelimumab (subjects with R/R HNSCC who have disease progression after receiving at least one platinum-based chemotherapy and at least one anti-PD-1/PD-L1) SoC treatment	ICOS CTLA-4	Phase: Phase II Intervention Model: Parallel assignment, randomized Outcome Measures: Number of subjects with dose-limiting toxicities Overall response rate Disease control rate PFS DoR Others	114
NCT02575404	GR-MD-02 Plus Pembrolizumab in Melanoma, Non-Small Cell Lung Cancer, and Squamous Cell Head and Neck Cancer Patients	Recruiting	GR-MD-02 in combination with standard Pembrolizumab treatment	Gal-3 /LAG3 axis PD-1	Phase: Phase Ib Intervention Model: Parallel assignment, non-randomized Outcome Measures: Frequency and severity of treatment-related adverse events measured by Common Terminology Criteria for Adverse Events (CTCAEs) Version 4.0 overall response rate Measure the response rate to combined therapy with GR-MD-02 and Pembrolizumab in patients Assess the biological activity of GR-MD-02 in combination with Pembrolizumab	22

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
