# Peer review of "Immune Checkpoints Pathways in Head and Neck Squamous Cell Carcinoma"

_cancers, 2021, doi:10.3390/cancers13051018_

Round 1

Reviewer 1 Report

This once revised manuscript addresses many concerns of the previous review and provides a comprehensive up to date summary of immune checkpoint pathways and related clinical trials. Given that immunotherapies have met with some success in the treatment of advanced HNSCC, a summary of different immunotherapy approaches, including combination treatments, is likely to be valuable to basic, translational and clinical investigators in the field. 

One concern remains about English, as the manuscript requires additional editing for grammar and style.

Author Response

We thank the reviewers for their comments on our review article. In addressing their comments as outlined below, we believe we have strengthened the manuscript. Please note that all new changes are presented in the main text in red color. We hope the reviewers find our revised version suitable for publication.

Reviewer: 1
Comments to the Author and Suggestions for Authors:

This once revised manuscript addresses many concerns of the previous review and provides a comprehensive up to date summary of immune checkpoint pathways and related clinical trials. Given that immunotherapies have met with some success in the treatment of advanced HNSCC, a summary of different immunotherapy approaches, including combination treatments, is likely to be valuable to basic, translational and clinical investigators in the field.

One concern remains about English, as the manuscript requires additional editing for grammar and style.

Response: We thank the reviewer for this observation. We have thoroughly edited the manuscript for grammar and style.

Reviewer 2 Report

All the questions were answered by the Authors.

Author Response

Reviewer: 2

Comments to the Author and Suggestions for Authors:

All the questions were answered by the Authors.

Response: We are happy to have answered all his/her questions.

Reviewer 3 Report

Review on immunotherapy of HNSCC.

The review focuses on the ongoing trials in HNSCC concerning immune checkpoint inhibitors. There molecular effects are summarized in 1 drawing and the trials in 1 table.

Please find below my comments on the manuscript:

The colours chosen in the drawing makes it difficult to rapidly identify the main message to show how the checkpoint inhibitors act. Overall the picture likes nice and also illustrates the involved pathway in HNSCC.

Few punctuation and typing errors: e.g.: “in accordance with the phenotype exhibit by T cell in HNSCC patients’ tumor biopsies [58].,Further studies” or “diminished IFN- signaling”

Concerning HNSCC as a clinical entity there are a lot of imprecisions and errors that should absolutely be addressed. A lot of rewriting is needed in the introduction part.

"During the last decades, the understanding of tumors’ immune profile became more relevant due to the development of novel immunotherapeutic strategies, especially immune checkpoint blockers." The sentence would be more meaningful the other way around: The research in the area of immunoncology und better understanding of the pathophysiology lead to new drugs such as the checkpoint inhibitors.  

"The blockade of PD-1 by monoclonal antibodies is the only immunotherapy approved for head and neck squamous cell carcinoma."  This sentence is not completely correct if you think about EGFR-inhibitors.

"Squamous cell carcinoma arises from the proliferation of squamous cells found in the outer layer of the skin and the mucous membranes." This sentence is misleading I think, since you refer mainly to HNSCC and not skin cancer in the head and neck area. Otherwise it would be helpful to specifiy more clearly.

"HNSCC develops in the mucous membranes of the oropharyngeal and larynx, oral cavity, hypopharynx, nasopharynx, and nasal cavity." In all anatomic areas that are covered with mucous skin HNSCC is arising. Not only nasal cavity, but also sinuses are part of it.

"Epidemiologically, HNSCC constitutes approximately 4% of all cancers worldwide [1]." This sentence suggests HNSCC to be a rare disease, although it is counted to be within the sixth most common cancers worldwide.

"Around 65.630 new cases of 58 oral cavity, pharyngeal and laryngeal cancer types will occur in 2020 59 with an estimated number of deaths of 14.500 patients [2]." These numbers seems to be up-to-date, but please read the limitations from the paper:

Note:

  • These are model‐based estimates that should be interpreted with caution and not compared with those for previous years.

Just by calculating 4% of around 1.8million cancers worldwide yields a very different number that is usually published. I am not sure the indicated incidence rate is adequate.

"The initial diagnosis of HNSCC is often clinical and is based on the patient’s symptoms that vary from chronic pain in the throat, tongue, or mouth, to hoarseness, difficulty swallowing, irregularly mass or ulcerations. Pathological confirmation of the diagnosis by analysis of tumor biopsy is mandatory [8]." Apart from the mentioned symptoms, a major form of how the cancer diagnosis is made is by lump in the neck!!!

"The increasing incidence of HPV-associated HNSCC and its relevant role as a prognostic biomarker in oropharyngeal squamous cell carcinomas (OSCC) supports the molecular testing for HPV infection in all patients diagnosed with this HNSCC subtype. In this regard, current guidelines indicate the pathological evaluation of HPV only in OSCC by p16 immunohistochemistry (IHC). In case of unknown primary HNSCC or unclear scenarios the current recommendation is to test p16 by IHC and then for p16+ tumors, HPV-specific detection assays [8,10]." Please rephrase. Complicated. Current guidelines request p16 IHC in OSCC and in the lymph nodes in CUP Syndrom. I do not understand what is an unclear scenario? If available HPV-specific detection can be performed to prove HPV as an underlying disease. 

" The NCCN and ESMO guidelines report that early-stage tumors are recommended to be treated by a single arm-modality with either surgery or radiotherapy (RT) without differences in survival rates.” There is an ongoing debate about this topic. In certain conditions surgery is better and the other way around. I would be careful to put it like this.

“Particularly, the treatment is uniform within each HNSCC entity, being surgery usually preferred for oral cavity and RT for nasopharyngeal carcinomas despite no differences in survival [8,10].” Again same topic. Modality may differ a lot in morbidity and depending on the situation also the survival. Surgery is usually used for oral cavity, whereas RT in NPC, yes.

“Generally, surgery is preferred for the oral cavity followed by RT or CRT while for other sites, surgery is reserved for smaller tumors whose response to induction CRT was poor.” Be careful. This statement is of ongoing debate and each centre differs a lot on how to treat each cancer. Looking at the NCCN guidelines for example, there are very vague about which modality to choose. Each patient is treated after presentation at the tumor board depending on several aspects. Thinking about a bigger laryngeal cancer with cartilage involvement, surgery is often the treatment Nr.1!!!

“However, around 50% of patients with locally advanced HNSCC develop recurrence after primary treatment with metastatic, local or regional disease and have poor prognosis with a median overall survival (OS) under 12 months (6 to 9 months) [11].” Misleading! Without treatment OS is below 12months, to my best knowledge higher with treatment.

“The treatment of these patients or those with metastatic disease at initial presentation (5-10%) is dictated by the patient’s performance status and intent of treatment (i.e., palliative vs. curative)” Can you talk about curative treatment in a patient with distant metastatic disease. NO!!!

“Patients with locoregional recurrence are candidates for surgical salvage, and those patients not amenable to surgery are candidates for curative-intent RT or reirradiation with or without chemotherapy. On the other hand, for patients with recurrent or metastatic (R/M) HNSCC not amenable to the mentioned curative options, palliative systemic therapy is indicated [12].” These clinical statements are very implicative and not 100% accurate. Treatment of patients with locoregional recurrence are complex and the decision on how to treat depends on multiple factors.

“and novel predictive biomarkers of response to anti-PD-1 therapy.” Better anti-PD-1/PD-L1 therapy.

Concerning chapter 5.2. I would add the data from the Ang study that you mention as (6). The mechanism you explain concerning tobacco smoke and DNA damages seem to be the reason why HPV+ smokers have a worse OS compared to HPV+ never smokers.

Concerning chapter 5.6. Apart from the biomarkers you mention, there are other biomarkers of hope: exosomes, CTCs and tumorDNA. This could maybe be mentioned.

Conclusion.

“The only immunotherapeutic approach approved to treat HNSCC involves inhibition of the immune checkpoint pathway PD-1/PD-L1 with the use of anti-PD-1 blockers.” Again EGFR-inhibitor omitted. Or formulate otherwise.

Author Response

Reviewer: 3

Comments to the Author and Suggestions for Authors:

The review focuses on the ongoing trials in HNSCC concerning immune checkpoint inhibitors. There molecular effects are summarized in 1 drawing and the trials in 1 table.

Please find below my comments on the manuscript:

  • The colours chosen in the drawing makes it difficult to rapidly identify the main message to show how the checkpoint inhibitors act. Overall the picture likes nice and also illustrates the involved pathway in HNSCC. Few punctuation and typing errors: e.g.: “in accordance with the phenotype exhibit by T cell in HNSCC patients’ tumor biopsies [58].,Further studies” or “diminished IFN- signaling”

Response: We thank the reviewer for the comment about the figure and for raising the punctuation point. We have amended the text accordingly.

  • Concerning HNSCC as a clinical entity there are a lot of imprecisions and errors that should absolutely be addressed. A lot of rewriting is needed in the introduction part.

Response: We thank the reviewer for this observation. We have thoroughly edited the manuscript following his/her insightful comments as detailed in the following points.

  • "During the last decades, the understanding of tumors’ immune profile became more relevant due to the development of novel immunotherapeutic strategies, especially immune checkpoint blockers." The sentence would be more meaningful the other way around: The research in the area of immunoncology und better understanding of the pathophysiology lead to new drugs such as the checkpoint inhibitors.

Response: We thank the reviewer for this observation. We have reformulated this sentence on page 1, lines 17-20.

  • "The blockade of PD-1 by monoclonal antibodies is the only immunotherapy approved for head and neck squamous cell carcinoma." This sentence is not completely correct if you think about EGFR-inhibitors.

Response: We thank the reviewer for this observation. As the reviewer correctly highlights, Cetuximab is a monoclonal antibody targeting EGFR approved for HNSCC treatment and could be considered an immunotherapeutic approach since, besides blocking the EGFR signaling, it has the ability to induce an immune response against tumor cells by ADCC. Therefore, we have reformulated this sentence on page 1, lines 20-22.

  • "Squamous cell carcinoma arises from the proliferation of squamous cells found in the outer layer of the skin and the mucous membranes." This sentence is misleading I think, since you refer mainly to HNSCC and not skin cancer in the head and neck area. Otherwise it would be helpful to specifiy more clearly.

Response: We thank the reviewer for this observation. We agree with him/her comment. We have replaced the sentence on page 2, lines 54-56.

  • "HNSCC develops in the mucous membranes of the oropharyngeal and larynx, oral cavity, hypopharynx, nasopharynx, and nasal cavity." In all anatomic areas that are covered with mucous skin HNSCC is arising. Not only nasal cavity, but also sinuses are part of it.  

Response: We thank the reviewer for this comment. We have amended the text accordingly on page 2, lines 56-58.

  • "Epidemiologically, HNSCC constitutes approximately 4% of all cancers worldwide [1]." This sentence suggests HNSCC to be a rare disease, although it is counted to be within the sixth most common cancers worldwide.

Response: We thank the reviewer for raising this point. As the reviewer highlights, HNSCC is one of the most common tumors diagnosed worldwide and its incidence continues to raise. We have amended the above mentioned sentence on page 2, lines 59-60. A reference was included in the reference list, accordingly.

  • "Around 65.630 new cases of oral cavity, pharyngeal and laryngeal cancer types will occur in with an estimated number of deaths of 14.500 patients [2]." These numbers seems to be up-to-date, but please read the limitations from the paper.

Response: We thank the reviewer for raising this point. In the current version we have updated the values to the worldwide 2020 estimation recently released by the public database “Global Cancer Observatory (GLOBOCAN)”. The updated version is now on Page 2, lines 60-62. A reference was included in the reference list, accordingly.

  • Just by calculating 4% of around 1.8million cancers worldwide yields a very different number that is usually published. I am not sure the indicated incidence rate is adequate.

Response: We thank the reviewer for raising this point. We have amended the text following his advice. Please see page 2, lines 59-62.

  • "The initial diagnosis of HNSCC is often clinical and is based on the patient’s symptoms that vary from chronic pain in the throat, tongue, or mouth, to hoarseness, difficulty swallowing, irregularly mass or ulcerations. Pathological confirmation of the diagnosis by analysis of tumor biopsy is mandatory [8]." Apart from the mentioned symptoms, a major form of how the cancer diagnosis is made is by lump in the neck!!!

Response: We thank the reviewer for this comment. We have amended the text accordingly, please see page 2, lines 75-81.

  • The increasing incidence of HPV-associated HNSCC and its relevant role as a prognostic biomarker in oropharyngeal squamous cell carcinomas (OSCC) supports the molecular testing for HPV infection in all patients diagnosed with this HNSCC subtype. In this regard, current guidelines indicate the pathological evaluation of HPV only in OSCC by p16 immunohistochemistry (IHC). In case of unknown primary HNSCC or unclear scenarios the current recommendation is to test p16 by IHC and then for p16+ tumors, HPV-specific detection assays [8,10]." Please rephrase. Complicated. Current guidelines request p16 IHC in OSCC and in the lymph nodes in CUP Syndrom. I do not understand what is an unclear scenario? If available HPV-specific detection can be performed to prove HPV as an underlying disease.

Response: We thank the reviewer for this comment and we apologize for not making this point clearer on the previous version of the manuscript. We have modified the text accordingly by indicating the two recommendentions for routine HPV detection in the setting of HNSCC. The updated version is presented on pages 2 and 3, lines 85-97. A reference was included in the reference list.

  • " The NCCN and ESMO guidelines report that early-stage tumors are recommended to be treated by a single arm-modality with either surgery or radiotherapy (RT) without differences in survival rates.” There is an ongoing debate about this topic. In certain conditions surgery is better and the other way around. I would be careful to put it like this.

Response: We thank the reviewer for this insightful observation. We have reformulated the idea to attend to the reviewer’s suggestion and followed the NCCN and ESMO recommendations for the clinical management of early-stage HNSCC tumors. The updated version is on page 3, lines 102-107.

  • “Particularly, the treatment is uniform within each HNSCC entity, being surgery usually preferred for oral cavity and RT for nasopharyngeal carcinomas despite no differences in survival [8,10].” Again same topic. Modality may differ a lot in morbidity and depending on the situation also the survival. Surgery is usually used for oral cavity, whereas RT in NPC, yes.

Response: We thank the reviewer for this insightful observation. We have amended the text accordingly. The updated version is on page 3, lines 107-113. We hope to have addressed the reviewer's concern.

  • “Generally, surgery is preferred for the oral cavity followed by RT or CRT while for other sites, surgery is reserved for smaller tumors whose response to induction CRT was poor.” Be careful. This statement is of ongoing debate and each centre differs a lot on how to treat each cancer. Looking at the NCCN guidelines for example, there are very vague about which modality to choose. Each patient is treated after presentation at the tumor board depending on several aspects. Thinking about a bigger laryngeal cancer with cartilage involvement, surgery is often the treatment Nr.1!!!

Response: We agree with the reviewer about this matter, and we have corrected the text accordingly. Specifically, we have highlighted the importance of the discussion of each HNSCC case by a multidisciplinary tumor board  to make a decision regarding the optimal treatment modality for the patient. The amended version is now presented on page 3, lines 119-128.

  • “However, around 50% of patients with locally advanced HNSCC develop recurrence after primary treatment with metastatic, local or regional disease and have poor prognosis with a median overall survival (OS) under 12 months (6 to 9 months) [11].” Misleading! Without treatment OS is below 12months, to my best knowledge higher with treatment.

Response: We thank the reviewer for this observation, we have indicated that the mentioned OS is for patients without treatment.

  • “The treatment of these patients or those with metastatic disease at initial presentation (5-10%) is dictated by the patient’s performance status and intent of treatment (i.e., palliative vs. curative)” Can you talk about curative treatment in a patient with distant metastatic disease. NO!!!

Response: The reviewer is right about this matter and we apoplogize for this oversight. We have deleted this sentence in the text accordingly. 

  • “Patients with locoregional recurrence are candidates for surgical salvage, and those patients not amenable to surgery are candidates for curative-intent RT or reirradiation with or without chemotherapy. On the other hand, for patients with recurrent or metastatic (R/M) HNSCC not amenable to the mentioned curative options, palliative systemic therapy is indicated [12].” These clinical statements are very implicative and not 100% accurate. Treatment of patients with locoregional recurrence are complex and the decision on how to treat depends on multiple factors.

Response: We thank the reviewer for this observation. As the reviewer correctly highlights, the treatment HNSCC patients with locoregional recurrence is complex and it differs between patients based on several factors including toxicities, prior treatment, anatomical site, performance status, and morbity, among other factors. Therefore, we have reformulated this sentence on page 3, lines 132-141.

  • “and novel predictive biomarkers of response to anti-PD-1 therapy.” Better anti-PD-1/PD-L1 therapy.

Response: We thank the reviewer for raising this point, we have amended the text accordingly.

  • Concerning chapter 5.2. I would add the data from the Ang study that you mention as (6). The mechanism you explain concerning tobacco smoke and DNA damages seem to be the reason why HPV+ smokers have a worse OS compared to HPV+ never smokers.

Response: We thank the reviewer for this valuable and interesting observation. We have added in the text a brief discussion of Ang and colleagues data (Reference number 7 in the text and reference list) regarding how smoking status negatively impacts on OS of HPV+ HNSCC patients. Please see page 32, lines 757-759.

  • Concerning chapter 5.6. Apart from the biomarkers you mention, there are other biomarkers of hope: exosomes, CTCs and tumorDNA. This could maybe be mentioned.

Response: We thank the reviewer for this suggestion. We have briefly discussed exosomes as non-invasive biomarkers with great potential in the setting of PD-L1 (section 5.3 of the manuscript, page 32 and 33, lines 788-793). We have now also mentioned ctDNA and CTCs as candidate biomarkers under current investigation in the conclusion section of the manuscript (please see page 35, lines 904-907).

  • “The only immunotherapeutic approach approved to treat HNSCC involves inhibition of the immune checkpoint pathway PD-1/PD-L1 with the use of anti-PD-1 blockers.” Again EGFR-inhibitor omitted. Or formulate otherwise.

Response: We thank the reviewer for this observation. We have amended the text accordingly. Please see page 35, lines 898-899.

Round 2

Reviewer 3 Report

Thank you for the nicely updated manuscript. From my point of view all the major especially clinical issues could be addressed and corrected in the text. Although it is extremely difficult to give a concise overview of this large topic, I think that the manuscript gives a good overview of the actual research topics in this field.

This manuscript is a resubmission of an earlier submission. The following is a list of the peer review reports and author responses from that submission.

Round 1

Reviewer 1 Report

The manuscript by Veigas et al. provides a review of immune checkpoint pathways in head and neck cancer, their roles in evasion of immune surveillance, therapeutic targets and predictive biomarkers. In addition, it includes a comprehensive summary of ongoing clinical trials with assessing the effectiveness of specific immune checkpoint blockades and their combinations, as well as combinations with chemotherapeutic agents and vaccines. For the most part, the literature cited is appropriate as is the discussion of cited findings that collectively provide an important and timely overview of head and neck cancer immunobiology and immunotherapies. Nonetheless, there are a few issues that should be addressed to make the review more aligned with advances in precision medicine strategies and novel technologies.

Main issues/suggestions:

  1. The manuscript would benefit from the inclusion of recent evidence regarding the effects of intracellular localization of PD-L1 on anti PD-1 immunotherapy (Gao, Y., et al., Nature Cell Biology, 2020).
  2. More discussion of immunotherapy in precision medicine should be included. Specifically, this review would be enhanced by a brief description of ongoing efforts to integrate several key prognostic biomarkers, such as TMB, PD-L1 expression, and inflammatory gene signature in 3D to characterize tumor immune features predictive of response to immunotherapy which in turn, can be used for patient stratification to improve clinical outcomes.
  3. In order to assure the timely nature of this review, it would be important to describe recent advances in technologies and data science, that are collectively being applied to improve clinical outcomes of immunotherapies. For instance, digital pathology coupled with AI is being used to assess the proximity of immune and tumor cells. Furthermore, immune checkpoint pathways can be visualized in real-time to assess response to targeted immunotherapy. Lastly, reverse translation approaches interrogating the underlying immunobiology of clinical findings are increasingly validated through the application of computational methodologies and large data sets.

Minor concern:

The manuscript should be edited for grammar and spelling.

Author Response

We thank the reviewers for their comments on our review article. In addressing their comments as outlined below, we have strengthened the manuscript and re-structured some sections. Particularly, we have modified the introduction in order to correct the clinical information accordingly to current guidelines and reviewer comments. We have improved the description of some topics and have also incorporated other relevant issues suggested by the reviewers. Please note that all new changes are presented in the main text in red color. We hope the reviewers find our revised version suitable for publication.

Reviewer: 1
Comments to the Author:

The manuscript by Veigas et al. provides a review of immune checkpoint pathways in head and neck cancer, their roles in evasion of immune surveillance, therapeutic targets and predictive biomarkers. In addition, it includes a comprehensive summary of ongoing clinical trials with assessing the effectiveness of specific immune checkpoint blockades and their combinations, as well as combinations with chemotherapeutic agents and vaccines. For the most part, the literature cited is appropriate as is the discussion of cited findings that collectively provide an important and timely overview of head and neck cancer immunobiology and immunotherapies. Nonetheless, there are a few issues that should be addressed to make the review more aligned with advances in precision medicine strategies and novel technologies.

Main issues/suggestions:

  1. The manuscript would benefit from the inclusion of recent evidence regarding the effects of intracellular localization of PD-L1 on anti PD-1 immunotherapy (Gao, Y., et al., Nature Cell Biology, 2020).

Response: We thank the reviewer for this comment. Following her/his advice, we have included the role of intracellular localization of PD-L1 in the context of anti-PD-1 immunotherapy from Gao et al, Nat Cell Biology 2020 as well as other relevant recent evidence related to the PD-1/PD-L1 axis in cancer immunity on page 8, lines 308-319. Two references were included in the reference list accordingly.

  1. More discussion of immunotherapy in precision medicine should be included. Specifically, this review would be enhanced by a brief description of ongoing efforts to integrate several key prognostic biomarkers, such as TMB, PD-L1 expression, and inflammatory gene signature in 3D to characterize tumor immune features predictive of response to immunotherapy which in turn, can be used for patient stratification to improve clinical outcomes.

Response: We thank the reviewer for this comment. To attend the reviewer suggestion, we have now added a new section entitled “5.6 Novel insights in biomarkers: combined and integrative strategies” within section 5 (“Biomarkers of response to immunotherapy in HNSCC”), where we discuss the relevance of combining several predictive or prognostic biomarkers to better select patient for immunotherapy as well as other promising biomarkers with high value in the inference of the complex TME and tumor heterogeneity.

  1. In order to assure the timely nature of this review, it would be important to describe recent advances in technologies and data science, that are collectively being applied to improve clinical outcomes of immunotherapies. For instance, digital pathology coupled with AI is being used to assess the proximity of immune and tumor cells. Furthermore, immune checkpoint pathways can be visualized in real-time to assess response to targeted immunotherapy. Lastly, reverse translation approaches interrogating the underlying immunobiology of clinical findings are increasingly validated through the application of computational methodologies and large data sets.

Response: We thank the reviewer for this observation. To address it, we now discuss the novel technologies being implemented in deciphering the TME in the new section entitled “5.6 Novel insights in biomarkers: combined and integrative strategies” within section 5 (“Biomarkers of response to immunotherapy in HNSCC”).

Minor concern:

The manuscript should be edited for grammar and spelling.

Response: We thank the reviewer for this observation. We have thoroughly edited the manuscript.

Reviewer 2 Report

The manuscript entitled “Immune checkpoints pathways in head and neck squamous cell carcinoma” is an interesting and well-written review that summarizes the recent knowledge about the immune escape mechanisms of cancer cells in Head and Neck Squamous Cell Carcinomas (HNSCC), focusing on the immune checkpoints. The study is on a timely subject in view of increasing interest about the identification of immune-related biomarkers that can help select suitable HNSCC patients who may obtain the most benefit from the new therapies. I suggest some minor revision to improve the paper:

  • Some acronyms are introduced several times in the text (e.g. PD-L1)
  • As the importance of the topic related to PD-L1, I recommend updating the literature.
  • In the paragraph 2.1 (“Composition and activation profile of immune cells in the TME”) the Authors mention the role of tumor-associated macrophages (TAMs) expression in HNSCC.
  • Only minor language corrections should be necessary (e.g. table in page 16: “PDL-1”).

Author Response

We thank the reviewers for their comments on our review article. In addressing their comments as outlined below, we have strengthened the manuscript and re-structured some sections. Particularly, we have modified the introduction in order to correct the clinical information accordingly to current guidelines and reviewer comments. We have improved the description of some topics and have also incorporated other relevant issues suggested by the reviewers. Please note that all new changes are presented in the main text in red color. We hope the reviewers find our revised version suitable for publication.

Comments to the Author:

The manuscript entitled “Immune checkpoints pathways in head and neck squamous cell carcinoma” is an interesting and well-written review that summarizes the recent knowledge about the immune escape mechanisms of cancer cells in Head and Neck Squamous Cell Carcinomas (HNSCC), focusing on the immune checkpoints. The study is on a timely subject in view of increasing interest about the identification of immune-related biomarkers that can help select suitable HNSCC patients who may obtain the most benefit from the new therapies. I suggest some minor revision to improve the paper:

  • Some acronyms are introduced several times in the text (e.g. PD-L1)

Response: We thank the reviewer for raising this point, we have amended the text accordingly.

  • As the importance of the topic related to PD-L1, I recommend updating the literature.

Response: We thank the reviewer for this observation. We have updated the literature related to the PD-1/PD-L1 axis in cancer immunity highlighting the intracellular role of PD-L1 expression and how the PD-1/PD-L1 axis could have an impact on the tumor-draining lymph nodes in addition to its role in the TME. Please see page 8, lines 308-319. Two references were included in the reference list accordingly.

  • In the paragraph 2.1 (“Composition and activation profile of immune cells in the TME”) the Authors mention the role of tumor-associated macrophages (TAMs) expression in HNSCC.

Response: We thank the reviewer for raising this point. Since TAMs are one of the most relevant cells in the TME mediating immune suppression and tumor immune escape and are involved in mechanisms of resistance to immunotherapeutic approach’s as well as in patients prognosis (Bruni, D.; Angell, H. K.; Galon, J. The Immune Contexture and Immunoscore in Cancer Prognosis and Therapeutic Efficacy. Nat. Rev. Cancer 2020. https://doi.org/10.1038/s41568-020-0285-7), we have summarized their role in HNSCC, particularly how TAMs impact on survival and metastatic features of these tumors. We hope to have addressed the reviewer's concern.

  • Only minor language corrections should be necessary (e.g. table in page 16: “PDL-1”).
    Response: We thank the reviewer for raising this point, we have amended the text accordingly.

Reviewer 3 Report

The manuscript deals with the immune-oncological pathways in HNSCC. It discusses the underlying molecular mechanism and focuses on the on-going trials as well as the approved immunotherapies to HNSCC.

Overall nicely written review, but with a lot of wrong clinical informations that do not adhere to the current applicable guidelines (for example see NCCN guidelines). These issues should be addressed and corrected.

Please allow me the following comments:

  • "Although the treatment algorithm is identical regardless of HPV status, clinical differences between HPV+ and HPV-negative (HPV-) patients are now being tested separately in several clinical trials to evaluate biological and treatment-related questions [6,8,9]" One of the main questions is the descalation of the standard therapy. Please be more specific.
  • "The initial diagnosis is often clinical, based on the observation of an irregularly infiltrating mass, with ulcerations, but should always be confirmed by biopsy to differentiate between a possible tumor and some inflammatory or infectious diseases, like tuberculosis and sarcoidosis, that can mimic carcinomas clinically [10]." This is NOT true...the clinical picture may vary a lot...often patients present with a painless lump in the neck as a first sign...also at the site of the primary cancer, the clinical picture may vary...exophytic, submucous or ulcerative pattern are well known...AND tuberculosis and sarcoidosis are rare differential diagnoses...more common are traumatic or infectious disease e.g. viral or bacterial infections

  • "One of the most reliable techniques to achieve a correct diagnosis of a primary HNSCC is a pathological analysis of the tumor histology as its microscopic appearance may vary as a function of tumor differentiation. " Sorry but this is THE Goldstandard. No cancer diagnosis without confirmed histopathological report!!!!!!! Before we do extended surgery or heavy chemoirradiation or immunotherapy we need to be absolutely sure.

  • "Indeed, current guidelines recommend routine HPV detection as part of the pathologic evaluation, particularly in oropharyngeal squamous cell carcinomas (OSCC)." The guidelines currently asks to perform HPV testing ONLY in orophayngeal cancers (OSCC)!!!

  • "The main recommendation is combining two different methodologies, commonly beginning with p16 immunocytochemistry and then for p16+ tumors, HPV-specific detection assays". This is really not routine, looking at the 8th TNM classification, in most centers only p16 is performed since it is a good surrogate marker, although HPV-specific assays would be more sure. In clinics p16/HPV status, as you correctly mentioned above in the manuscript, does currently not change the treatment choice/alogrhythm. Only of predictive use....
  • "Although the HNSCC subtypes are clinically, histologically, and molecularly distinct, they are treated uniformly and with limited success." Again I see your point, but uniform treatment NO, at least not in our center. I think you want to express that within a HNSCC entity the treatment is similar and more attention should be put onto the molecular differences. AN example : nasopharynx carcinoma is almost always treated by radiotherapiy whereas oral cavity HNSCC is primarly treated by surgery. Please formulate more clearly.

  • Concerning limited success in the previous sentence from your manuscript, this holds only true for advanced cancer stages, the early ones mostly have good overall prognosis
  • "The first treatment option for primary HNSCC is surgical resection at any anatomical localization." NO....I do NOT agree...see my example from before...this is not current guidelines....Of course there are a lot of ongoing debates about what treatment modality to use...another example is early laryngeal cancer: radiotherapy vs transoral surgery....
  • "When incomplete resection or relapse occurs, the indication is to continue with radiotherapy or systemic treatments, or both." NOT true....

  • "In the advanced HNSCC stage, the tumor is usually unresectable, and the alternative is radiotherapy or systemic treatments." NONSENSE....please review your criteria for unresectability....for example in an M1 situation, surgery is often feasible, but of little oncological value
  • "The treatment of patients with unresectable locoregional, persistent, recurrent, or metastatic HNSCC is dictated by the patient’s performance status and intent of treatment (i.e., palliative vs. curative) [16]." To me this sentence is strange....there are different clinical entities....you mainly are talking about recurrent and metastatic HNSCC, here yes the Extreme chemotherapy scheme is mainly used and newly ICB can be used...
  • "HNSCC is considered an immunogenic tumor as it is often accompanied by a prominent immune infiltrate [18]."  Please indicate the correct reference, the one cited is only about TILs being a prognostic factor in general in cancer, but not specific  in HNSCC.
  • "5.3. PD-L1 expression" Here is miss some sentences on the CPS and TPS in the histology that are often used to decide on choice of immuno-/chemotherapy
  •  Can the author say something about why the mono-immunotherapies are not working as well in HNSCC compared to other cancer such as melanoma? Maybe this could be included in the manuscript.

Author Response

We thank the reviewers for their comments on our review article. In addressing their comments as outlined below, we have strengthened the manuscript and re-structured some sections. Particularly, we have modified the introduction in order to correct the clinical information accordingly to current guidelines and reviewer comments. We have improved the description of some topics and have also incorporated other relevant issues suggested by the reviewers. Please note that all new changes are presented in the main text in red color. We hope the reviewers find our revised version suitable for publication.

Reviewer: 3
Comments to the Author:

The manuscript deals with the immune-oncological pathways in HNSCC. It discusses the underlying molecular mechanism and focuses on the on-going trials as well as the approved immunotherapies to HNSCC.

Overall nicely written review, but with a lot of wrong clinical informations that do not adhere to the current applicable guidelines (for example see NCCN guidelines). These issues should be addressed and corrected.

Please allow me the following comments:

  • "Although the treatment algorithm is identical regardless of HPV status, clinical differences between HPV+ and HPV-negative (HPV-) patients are now being tested separately in several clinical trials to evaluate biological and treatment-related questions [6,8,9]" One of the main questions is the descalation of the standard therapy. Please be more specific.

Response: We thank the reviewer for this observation. As the reviewer correctly highlights, one of the main questions concerning HPV+ patients is the de-escalation of standard therapy. Therefore, we have reformulated this sentence as follows on page 3 lines 116-122: “Concerning HPV status, despite its prognostic value in OSCC patients, it does not change the treatment algorithm [8,10].  However, clinical differences between HPV+ and HPV- patients are now being tested separately in several ongoing or just finished clinical trials to evaluate biological and treatment-related questions such as how HPV status impacts tumor response to treatment and de-escalation of the standard therapy for HPV+ tumors [16,17]. De-escalation aims to decrease toxicity and morbidity resulted from SoC while maintaining tumor control, quality of life, and favorable survival [17]. A reference was included in the reference list accordingly.

  • "The initial diagnosis is often clinical, based on the observation of an irregularly infiltrating mass, with ulcerations, but should always be confirmed by biopsy to differentiate between a possible tumor and some inflammatory or infectious diseases, like tuberculosis and sarcoidosis, that can mimic carcinomas clinically [10]." This is NOT true...the clinical picture may vary a lot...often patients present with a painless lump in the neck as a first sign...also at the site of the primary cancer, the clinical picture may vary...exophytic, submucous or ulcerative pattern are well known...AND tuberculosis and sarcoidosis are rare differential diagnoses...more common are traumatic or infectious disease e.g. viral or bacterial infections.

Response:  We agree with the reviewer, and we have corrected the text accordingly based on EHNS-ESMO-ESTRO Clinical Practice Guidelines for diagnosis, treatment, and follow up. This is reflected on page 2, lines 63-65. A reference was included in the reference list accordingly.

  • "One of the most reliable techniques to achieve a correct diagnosis of a primary HNSCC is a pathological analysis of the tumor histology as its microscopic appearance may vary as a function of tumor differentiation. " Sorry but this is THE Goldstandard. No cancer diagnosis without confirmed histopathological report!!!!!!! Before we do extended surgery or heavy chemoirradiation or immunotherapy we need to be absolutely sure.

Response: We thank the reviewer for raising this point and we apologise for not making this point clearer. We have modified the text accordingly by indicating on page 2, lines 65-66 that “Pathological confirmation of the diagnosis by analysis of tumor biopsy is mandatory”.

  • "Indeed, current guidelines recommend routine HPV detection as part of the pathologic evaluation, particularly in oropharyngeal squamous cell carcinomas (OSCC)." The guidelines currently asks to perform HPV testing ONLY in orophayngeal cancers (OSCC)!!!

Response: The reviewer is right about this matter and we apoplogise for this oversight. We have amended the text in accordance to current guidelines highlighting that HPV testing, as the reviewer correctly mentiona, is only requested in OSCC patients (i.e. EHNS-ESMO-ESTRO Clinical Practice Guidelines and NNCN Guidelines Version 1.2021). Please see page 2, lines 68-72: “The increasing incidence of HPV-associated HNSCC and its relevant role as a prognostic biomarker in oropharyngeal squamous cell carcinomas (OSCC) supports the molecular testing for HPV infection in all patients diagnosed with this HNSCC subtype. In this regard, current guidelines indicate the pathological evaluation of HPV only in OSCC by p16 immunohistochemistry (IHC)… [8,10]”. A reference was included in the reference list accordingly and the NCCN guidelines have been updated to the last version.

  • "The main recommendation is combining two different methodologies, commonly beginning with p16 immunocytochemistry and then for p16+ tumors, HPV-specific detection assays". This is really not routine, looking at the 8th TNM classification, in most centers only p16 is performed since it is a good surrogate marker, although HPV-specific assays would be more sure. In clinics p16/HPV status, as you correctly mentioned above in the manuscript, does currently not change the treatment choice/alogrhythm. Only of predictive use....

Response: We thank the reviewer for this remarkable comment. As the reviewer correctly highlights, p16 IHC is a good surrogate marker for HPV infection being this the recommended HPV identification method for patients diagnosed with OSCC. However, a few other testing options are available for use in the clinical setting such as PCR and in situ hybridization (ISH). Particularly for unclear scenarios, an equivocal p16, or patients diagnosed with neck metastasis for unknown origin, multiple methods may be used to confirm HPV status. Specifically, PCR and ISH will provide additional sensitivity and specificity (NNCN Guidelines Version 1.2021 and EHNS-ESMO-ESTRO Clinical Practice Guidelines). In this regard, we have modified the text on page 2, lines 72-74 as follows: “…In case of unknown primary HNSCC or unclear scenarios the current recommendation is to test p16 by IHC and then for p16+ tumors, HPV-specific detection assays [8,11].”  A reference was included in the reference list accordingly and the NCCN guidelines had been updated to the latest version.

  • "Although the HNSCC subtypes are clinically, histologically, and molecularly distinct, they are treated uniformly and with limited success." Again I see your point, but uniform treatment NO, at least not in our center. I think you want to express that within a HNSCC entity the treatment is similar and more attention should be put onto the molecular differences. AN example: nasopharynx carcinoma is almost always treated by radiotherapiy whereas oral cavity HNSCC is primarly treated by surgery. Please formulate more clearly.

Response: We thank the reviewer for raising this point. We have re-formulated the sentence in the text based on current guidelines for diagnosis, treatment, and follow up on pages 2, lines 76-82. A reference was included in the reference list accordingly and the NCCN guidelines had been updated to the last version. We hope we have successfully addressed the reviewer's concern.

  • Concerning limited success in the previous sentence from your manuscript, this holds only true for advanced cancer stages, the early ones mostly have good overall prognosis

Response: We agree with the reviewer observation, and we have amended the text accordingly indicating the differences in prognosis between HNSCC stages. Please see page 2, lines 82-92.

  • "The first treatment option for primary HNSCC is surgical resection at any anatomical localization." NO....I do NOT agree...see my example from before...this is not current guidelines....Of course there are a lot of ongoing debates about what treatment modality to use...another example is early laryngeal cancer: radiotherapy vs transoral surgery....

Response: We thank the reviewer for this observation. We have amended the text accordingly to current guidelines for early stage HNSCC tumors. Please see page 2, lines 78-82. A reference was included in the reference list accordingly and the NCCN guidelines had been updated to the last version.

  • "When incomplete resection or relapse occurs, the indication is to continue with radiotherapy or systemic treatments, or both." NOT true....

Response: We thank the reviewer for raising this point and we have modified the text to better represent the therapeutic options. Particularly, we have indicated the therapeutic options for locally advanced HNSCC treatment and for patients who develop recurrence after these primary treatments. Please see pages 2 and 3, lines 85-101.

  • "In the advanced HNSCC stage, the tumor is usually unresectable, and the alternative is radiotherapy or systemic treatments." NONSENSE....please review your criteria for unresectability....for example in an M1 situation, surgery is often feasible, but of little oncological value.

Response: We thank the reviewer for this observation and we fully agree that unresectability and inoperable are different concepts. We have re-written most of this section and added a more in detail description of the different therapeutic options for each HNSCC stage. We hope to have properly addressed the reviewer’s concern.

  • "The treatment of patients with unresectable locoregional, persistent, recurrent, or metastatic HNSCC is dictated by the patient’s performance status and intent of treatment (i.e., palliative vs. curative) [16]." To me this sentence is strange....there are different clinical entities....you mainly are talking about recurrent and metastatic HNSCC, here yes the Extreme chemotherapy scheme is mainly used and newly ICB can be used...

Response: We thank the reviewer for raising this point. Based on current guidelines, the treatment of patients with locally advanced disease who develop recurrence to treatment (local, regional, or metastatic), as well as for those with metastatic disease at the time of diagnosis, is defined by performance status and intent of treatment. To address this comment, we have reformulated the text on page 2 and 3, lines 92-111.

  • "HNSCC is considered an immunogenic tumor as it is often accompanied by a prominent immune infiltrate [18]." Please indicate the correct reference, the one cited is only about TILs being a prognostic factor in general in cancer, but not specific in HNSCC.

Response: We thank the reviewer for this observation. We have changed the reference in the text to one more specific concerning the immune landscape of HNSCC tumors: Mandal, R.; Şenbabaoğlu, Y.; Desrichard, A.; Havel, J. J.; Dalin, M. G.; Riaz, N.; Lee, K.-W.; Ganly, I.; Hakimi, A. A.; Chan, T. A.; et al. The Head and Neck Cancer Immune Landscape and Its Immunotherapeutic Implications. JCI insight 2016, 1 (17), e89829–e89829. https://doi.org/10.1172/jci.insight.89829 (reference number 21 in the text).

  • PD-L1 expression" Here is miss some sentences on the CPS and TPS in the histology that are often used to decide on choice of immuno-/chemotherapy

Response: We thank the reviewer for this observation regarding the usefulness of PD-L1 expression as prognostic biomarker in clinical management of HNSCC. To address this point, we have re-writen this sub-section (5.3 PD-L1 expression) within the section 5 (“Biomarkers of response to immunotherapy in HNSCC”) to further discuss the PD-L1 CPS and TPS scores in HNSCC. Please see pages 27 and 28.

  • Can the author say something about why the mono-immunotherapies are not working as well in HNSCC compared to other cancer such as melanoma? Maybe this could be included in the manuscript.

Response: We thank the reviewer for raising this point. We think that it is beyond of the scope this review to discuss the specific molecular mechanisms of the limited clinical benefit of anti PD-1 monotherapy in HNSCC.

Reviewer 4 Report

Peer review report

Manuscript title: “Immune checkpoints pathways in head and neck  squamous cell carcinoma”

Authors: Florencia Veigas, Adriana Rinflerch, Yamil D. Mahmoud, Joaquin Merlo, Gabriel A.  Rabinovich, and María Romina Girotti.

Word count: approx. 7000

I congratulate the authors on their efforts and manuscript, in which they aim to give an overview of head and neck cancer immunology and immunotherapy. To do so, they have written a narrative review describing the tumor microenvironment, immune checkpoints, trials investigating immune-checkpoint blockade in HNSCC and biomarkers for immunotherapy response in HNSCC.

I applaud the authors’ attempt to produce an extensive review, though at approximately 7000 words it is rather lengthy. But more importantly, I find the manuscript not concisely written and unfocused. I think the paper would benefit from a clearer structure and more carefully chosen aims or focus areas. It is, for example, currently unclear why a lengthy paragraph is dedicated to Galectin-1, while the effects of TGF-beta, IL-10, IL-6, PGE2 and VEGF are briefly summarized in only one (much shorter) paragraph. In addition, the combination of immunotherapy with (chemo)radiotherapy is extensively discussed, while recent data on neoadjuvant immune checkpoint blockade prior to surgery are omitted and CAR-T cells are briefly mentioned for the first time in the final, concluding paragraph.

In addition, I think the manuscript is not well-structured. In the introduction, for example, the authors move from etiology and diagnosis to biomarkers, genetic landscape, treatment, back to genetic landscape and then back to treatment. It would greatly help the reader to have a more logically structured introduction, and to have it end with a statement on what the authors intend to review, which is now completely unclear. Also,  the authors need to present the data from the studies they discuss in the main text section in a more logical order. In this manuscript, the authors move back and forth between human, mouse and in vitro studies. This makes it very hard for me to appreciate the value, weight and potential clinical applicability of each finding. Examples are: lines 190-206, 269-281 and 318-330.

Finally, I find the authors’ description of the available literature mostly superficial, often vague and at times unscientific or even incorrect.  While the narrative review format leaves the authors some freedom to summarize for the sake of readability, I think they have done so too much in this manuscript. This leaves me wonder how well the authors know the presented literature. Some examples are:

  • However, they have a modest prognostic impact in part due to the complexity of this disease” (line 68-69).
  • Although the HNSCC subtypes are clinically, histologically, and molecularly distinct, they are treated uniformly and with limited success” (lines 83-84). What HNSCC subtypes do you mean? Also, appropriate treatment is determined according to disease site and stage and patient factors. Not all HNSCC are therefore treated uniformly. “Limited success” is not scientific language.
  • “The first treatment option for primary HNSCC is surgical resection at any anatomical localization. When incomplete resection or relapse occurs, the indication is to continue with radiotherapy or systemic treatments, or both. In the advanced HNSCC stage, the tumor is usually unresectable, and the alternative is radiotherapy or systemic treatments.” (lines 86-89). This is incorrect. Depending on site and stage, definitive (chemo)radiotherapy may be the preferred curative treatment option. (Chemo)Radiotherapy is not only reserved for patients with disease relapse. Also, some patients with disease recurrence may be surgically salvaged.
  • Teff cells” (used multiple times throughout the manuscript): I find this unspecific and unclear. What do you mean with effector T cells? These could still have many phenotypes.
  • These patients have a poor prognosis with a median overall survival (OS) under 12 months” (103-104). This could be anything ranging from 1 day to 11.9 months.
  • Although LAG-3 functions in coordination with other checkpoint molecules to promote T-cell dysfunction, the molecular mechanisms and pathways implicated in LAG-3 signaling are still under scrutiny.“ (line 309-311)
  • Restoring the immune response” (line 338).
  • In this context, TIM-3-expressing Tregs infiltrating HNSCC mediates resistance to immunotherapy in particular by the immunosuppressive effects that triggers in the TME.” (Lines 339-341).
  • Besides Tregs, the pro-tumorigenic and immunosuppressive M2-like macrophages are related to the TIM-3/Gal-9 pathway” (345-346).
  • …clinical benefit on PFS and objective response” (line 378)
  • Later, in 2019, based on KEYNOTE-048 results, 379 Pembrolizumab has been approved as monotherapy by FDA for first-line treatment of patients with 380 PD-L1 positive R/M HNSCC or in combination with platinum and 5-fluorouracil in R/M HNSCC. In 381 particular, KEYNOTE-048, a randomized, open-label phase 3 study showed that Pembrolizumab 382 alone or with chemotherapy improved median response duration by more than 16 and 2.5 months, 383 respectively, versus cetuximab with chemotherapy. Moreover, Pembrolizumab monotherapy’s 384 profound OS benefits were observed in patients with PDL1positive tumors while for 385 Pembrolizumab with chemotherapy in all participants.” (lines 379-386). One of the major conclusions of the KEYNOTE-048 trial is that patients with PD-L1 CPS greater than 1 (but not the total population) benefit from Pembrolizumab monotherapy. Patients with CPS <1 may benefit from pembro + chemotherapy, yet this regimen is very toxic. I do not think that these important conclusions are adequately drawn in this paragraph.
  • The authors refer to the phase III JAVELIN-100 and IMvoke010 trials as testing “safety and anti-tumor efficacy” (lines 387-388) and “efficacy and safety” (line 392), respectively, of different immunotherapy regimens combined with (chemo)radiotherapy. While safety and toxicity are of course registered in phase III trials, they are never primary endpoints. For example, IMvoke010’s primary endpoint is PFS.
  • Although no statistically significant differences in OS were found for the monotherapy or the combination versus SoC, higher survival, and response rates at 12 and 24 months demonstrated the clinical activity of durvalumab” (lines 403-405). This is a near-exact copy of the original EAGLE trial’s abstract conclusion: “There were no statistically significant differences in OS for durvalumab or durvalumab plus tremelimumab versus SoC. However, higher survival rates at 12 to 24 months and response rates demonstrate clinical activity for durvalumab”.
  • PD-L1 expression on tumor and immune cells has been associated with improved treatment outcomes in several cancers” (lines 522-523).
  • Interestingly, intra-tumor heterogeneity is discussed as a potential biomarker of response to immunotherapy based on its impact on the immune profile of the TME and association with survival” (lines 529-531).
  • In a scRNA-Seq study of primary tumors and lymph node metastasis biopsies from HNSCC patients , the immune types found varied in their proportions. However, they showed consistent expression profiles across patients.” (lines 562-564).

In conclusion, the lack of focus, structure and sufficiently in-depth reporting on the available literature leads me to regard the present manuscript as unsuitable for publication.

Author Response

We thank the reviewers for their comments on our review article. In addressing their comments as outlined below, we have strengthened the manuscript and re-structured some sections. Particularly, we have modified the introduction in order to correct the clinical information accordingly to current guidelines and reviewer comments. We have improved the description of some topics and have also incorporated other relevant issues suggested by the reviewers. Please note that all new changes are presented in the main text in red color. We hope the reviewers find our revised version suitable for publication.

Reviewer: 4
Comments to the Author:

I congratulate the authors on their efforts and manuscript, in which they aim to give an overview of head and neck cancer immunology and immunotherapy. To do so, they have written a narrative review describing the tumor microenvironment, immune checkpoints, trials investigating immune-checkpoint blockade in HNSCC and biomarkers for immunotherapy response in HNSCC.

I applaud the authors’ attempt to produce an extensive review, though at approximately 7000 words it is rather lengthy. But more importantly, I find the manuscript not concisely written and unfocused. I think the paper would benefit from a clearer structure and more carefully chosen aims or focus areas. It is, for example, currently unclear why a lengthy paragraph is dedicated to Galectin-1, while the effects of TGF-beta, IL-10, IL-6, PGE2 and VEGF are briefly summarized in only one (much shorter) paragraph. In addition, the combination of immunotherapy with (chemo)radiotherapy is extensively discussed, while recent data on neoadjuvant immune checkpoint blockade prior to surgery are omitted and CAR-T cells are briefly mentioned for the first time in the final, concluding paragraph.

Response: We thank the reviewer for this comment. We have endeavoured to improve the manuscript structure and to deepen the focus of this review.  Regarding Galectin-1, we have summarized its role in HNSCC immune scape in a shorter paragraph than the oiriginally submitted version. We consider Galectin-1 of significant relevance due to the increasing evidence of this soluble molecule as a modulator of the immune response in several tumor types including HNSCC as well as it impacts in response and resistance mechanisms to anti-PD-1 immunotherapy. In response to reviewer’s comment on neoadjuvant immune checkpoint blockade we have added a new subsection (4.3) entitled “ICB in the neoadjuvant setting”. To improve the structure of the manuscript we now discuss on this section two clinical trials (IMvoke10 and IMSTAR-HN) that in the originally submitted version of the manuscript were described under the section 4. Concerning CAR-T cells, as the reviewer indicates in him/her comment, they are only mentioned in the concluding paragraph of the manuscript as a future promising direction in the treatment of HNSCC tumors. To date, CAR-T cells have demonstrated to be successful in eradicating hematologic malignancies like acute and chronic B cell leukemias. However, there are many factors that limit the use of CAR T cells in solid tumors such as the immunosuppressive TME and the identification of specific tumor-associated antigens (June, C. H.; O’Connor, R. S.; Kawalekar, O. U.; Ghassemi, S.; Milone, M. C. CAR T Cell Immunotherapy for Human Cancer. Science 2018, 359 (6382), 1361–1365. https://doi.org/10.1126/science.aar6711). In this regard, to our knowledge there is only one clinical trial testing the efficacy of an intra-tumoral CAR-T cell immunotherapy in the setting of locally advanced and recurrent HNSCC. We now discuss on the manuscript text this ongoing clinical trial and mentioned a recent study in which Mucine-1 is proposed as promising target for CAR-T cells.  The new references have been included in the reference list, accordingly.

In addition, I think the manuscript is not well-structured. In the introduction, for example, the authors move from etiology and diagnosis to biomarkers, genetic landscape, treatment, back to genetic landscape and then back to treatment. It would greatly help the reader to have a more logically structured introduction, and to have it end with a statement on what the authors intend to review, which is now completely unclear. Also,  the authors need to present the data from the studies they discuss in the main text section in a more logical order. In this manuscript, the authors move back and forth between human, mouse and in vitro studies. This makes it very hard for me to appreciate the value, weight and potential clinical applicability of each finding. Examples are: lines 190-206, 269-281 and 318-330.

Response: We thank the reviewer for this comment. We have rewritten the introduction section in a more structured way moving from etiology and genomic landscape to diagnosis, biomarkers and treatment. In addition, we have included a statement at the end of the introduction section summarizing what we intend to review in this manuscript. Attending the suggestion of a more logical order in the discussion of studies of Gal-1 (lines 190-206 of the originally submitted manuscript, now discussed on pages 5, lines 212-226), CTLA-4 (lines 269-281 of the originally submitted manuscript; now discussed on pages 7 and 8 lines 280-297), and LAG-3 (lines 318-330 of the originally submitted manuscript; now discussed on pages 9, lines 355-369).

Finally, I find the authors’ description of the available literature mostly superficial, often vague and at times unscientific or even incorrect.  While the narrative review format leaves the authors some freedom to summarize for the sake of readability, I think they have done so too much in this manuscript. This leaves me wonder how well the authors know the presented literature. Some examples are:

  • “However, they have a modest prognostic impact in part due to the complexity of this disease” (line 68-69).

Response: We thank the reviewer for this observation. In the current version of the manuscript, we have modified the introduction to address all the reviewer’s comments regarding clinical and structural aspects of this section. Thus, this sentence has been deleted.

  • “Although the HNSCC subtypes are clinically, histologically, and molecularly distinct, they are treated uniformly and with limited success” (lines 83-84). What HNSCC subtypes do you mean? Also, appropriate treatment is determined according to disease site and stage and patient factors. Not all HNSCC are therefore treated uniformly. “Limited success” is not scientific language.

Response: We thank the reviewer for this observation. To address this point, we have re-formulated the idea in the text based on clinical current guidelines highlighting that treatment options will vary with stage, anatomical site, patients’ intrinsic factors and accessibility as well as the relevance of a multidisciplinary team at the time of taking decision regarding treatment. The revised text is on page 2, lines 76-78.

We have also provided an overview of the treatment opportunities for each HNSCC stage. Particularly, we have indicated that within each HNSCC subtype (based on the site of the disease) the treatment modality is uniform choosing between surgery or radiotherapy as single-arm modality and the same for locally advanced HNSCC. Page 2, lines 78-82: “The NCCN and ESMO guidelines report that early-stage tumors are recommended to be treated by a single arm-modality with either surgery or radiotherapy (RT) without differences in survival rates. Particularly, the treatment is uniform within each HNSCC entity, being surgery usually preferred for oral cavity and RT for nasopharyngeal carcinomas despite no differences in survival [8,10].” One reference was included in the reference list accordingly and the NCCN guidelines had been updated to the last version. We have also added on page 2, lines 85-88: For locoregional advanced HNSCC, standard treatment often includes surgery plus adjuvant RT or chemoradiotherapy (CRT) or primary CRT alone. Generally, surgery is preferred for the oral cavity followed by RT or CRT while for other sites, surgery is reserved for smaller tumors whose response to induction CRT was poor... [8,10]”

  • “The first treatment option for primary HNSCC is surgical resection at any anatomical localization. When incomplete resection or relapse occurs, the indication is to continue with radiotherapy or systemic treatments, or both. In the advanced HNSCC stage, the tumor is usually unresectable, and the alternative is radiotherapy or systemic treatments.” (lines 86-89). This is incorrect. Depending on site and stage, definitive (chemo)radiotherapy may be the preferred curative treatment option. (Chemo)Radiotherapy is not only reserved for patients with disease relapse. Also, some patients with disease recurrence may be surgically salvaged.

Response: We thank the reviewer for this observation. As mentioned in the previous comment, we have re-formulated the text based on the current guidelines. These two sentences have been replaced on page 2 and 3 for lines 82-111 to provide a deeper description of the different therapeutic options for each HNSCC stage and mention, as the reviewer correctly indicates, that at the time of recurrence or relapse based on performance status, localization and intent of treatment, the options could vary. One reference was included in the reference list accordingly.

  • “Teff cells” (used multiple times throughout the manuscript): I find this unspecific and unclear. What do you mean with effector T cells? These could still have many phenotypes.

Response: We thank the reviewer for this observation. The term Teff used many times throughout this review article, refers to a group of activated T cells able to induce an effective anti-tumor immune response, either CD8 or CD4 T cells. This acronym is widely used in the context of tumor biology and immunotherapy to mention T cell with effector functions in general, indeed it was used in other published manuscripts such as Spranger S. Mechanisms of tumor escape in the context of the T-cell-inflamed and the non-T-cell-inflamed tumor microenvironment. Int Immunol. 2016 Aug;28(8):383-91. doi: 10.1093/intimm/dxw014 or Rotte A, Jin JY, Lemaire V. Mechanistic overview of immune checkpoints to support the rational design of their combinations in cancer immunotherapy. Ann Oncol. 2018 Jan 1;29(1):71-83. doi: 10.1093/annonc/mdx686. We hope the reviewer finds our explanation sufficient.

  • “These patients have a poor prognosis with a median overall survival (OS) under 12 months” (103-104). This could be anything ranging from 1 day to 11.9 months.

Response: The reviewer is right about this matter, and we are thankful for this comment. To be clearer and more specific, we have indicated in the manuscript the median OS for these patients (page 2, line 92).

  • “Although LAG-3 functions in coordination with other checkpoint molecules to promote T-cell dysfunction, the molecular mechanisms and pathways implicated in LAG-3 signaling are still under scrutiny.“ (line 309-311).

Response: We thank the reviewer for this comment. We have now mentioned the mechanisms know to be implicated in LAG-3 signaling until now on page 9, lines 347-354. Two references were included in the reference list accordingly.

  • “Restoring the immune response” (line 338)

Response: We thank the reviewer for raising this point. We have now mentioned the changes that occur in the immune compartment by the in vivo inhibition of TIM-3 in a preclinical mouse model of HNSCC triggering an effective anti-tumor immune response able to control tumor growth (Page 9, lines 384-387)

  • “In this context, TIM-3-expressing Tregs infiltrating HNSCC mediates resistance to immunotherapy in particular by the immunosuppressive effects that triggers in the TME.” (Lines 339-341).

Response: We thank the reviewer for this observation. We have re-formulated the idea on page 10, lines 388-393 as follows: Interestingly, in two orthotopic mouse models of HNSCC, the treatment with RT and anti-PD-1 induced the upregulation of TIM-3 in CD8 and CD4 T cells, mainly in Tregs, mediating resistance to treatment. Although targeting TIM-3 in addition to PD-L1 and RT results in tumor growth delay and improved survival through reducing intra-tumoral Tregs frequency, the response is not durable since remaining infiltrating Tregs are highly proliferative and could expand, fostering tumor recurrence [33]”. We have also have discussed the result of the studies related to TIM-3 in more detail (page 10, lines 376-394).

  • “Besides Tregs, the pro-tumorigenic and immunosuppressive M2-like macrophages are related to the TIM-3/Gal-9 pathway” (345-346).

Response: We thank the reviewer for raising this point, as we have re-written this section to be clearer and discuss the studies in a logical order, this sentence has been modified, on page 9, lines 380-381: “In this context, TIM-3/Gal-9 expression is closely associated with the Tregs and M2-like macrophages [86]”.

  • “…clinical benefit on PFS and objective response” (line 378)

Response: We thank the reviewer for raising this point. We have amended the text accordingly. Regarding PD-L1 expression, we have indicated the values for TPS and CPS associated with response to anti PD-1 therapy (1% and 50%, respectively; page 10, lines 427-430).

  • “Later, in 2019, based on KEYNOTE-048 results, 379 Pembrolizumab has been approved as monotherapy by FDA for first-line treatment of patients with 380 PD-L1 positive R/M HNSCC or in combination with platinum and 5-fluorouracil in R/M HNSCC. In 381 particular, KEYNOTE-048, a randomized, open-label phase 3 study showed that Pembrolizumab 382 alone or with chemotherapy improved median response duration by more than 16 and 2.5 months, 383 respectively, versus cetuximab with chemotherapy. Moreover, Pembrolizumab monotherapy’s 384 profound OS benefits were observed in patients with PDL1positive tumors while for 385 Pembrolizumab with chemotherapy in all participants.” (lines 379-386). One of the major conclusions of the KEYNOTE-048 trial is that patients with PD-L1 CPS greater than 1 (but not the total population) benefit from Pembrolizumab monotherapy. Patients with CPS <1 may benefit from pembro + chemotherapy, yet this regimen is very toxic. I do not think that these important conclusions are adequately drawn in this paragraph.

Response: We thank the reviewer for this interesting observation. We have modified the main text accordingly. Particular, we have now added the CPS information in the description of the clinical trials results on page 10 and 11, lines 435-437: Moreover, Pembrolizumab monotherapy’s profound OS benefits were observed only in patients with PD­L1­positive tumors (CPS ³ 1), while for Pembrolizumab with chemotherapy in all participants, regarding CPS”.

  • The authors refer to the phase III JAVELIN-100 and IMvoke010 trials as testing “safety and anti-tumor efficacy” (lines 387-388) and “efficacy and safety” (line 392), respectively, of different immunotherapy regimens combined with (chemo)radiotherapy. While safety and toxicity are of course registered in phase III trials, they are never primary endpoints. For example, IMvoke010’s primary endpoint is PFS.

Response: We thank the reviewer for this observation, we amended the text in accordance with reviewer comment. For JAVELIN-100 we have indicated that the primary outcome is PFS and we have also added the interim results published on September in ESMO. The current version in page 11, lines 440-445 reads: “JAVELIN 100 is a randomized, double-blind, phase III clinical trial testing safety and anti-tumor efficacy of Avelumab, a PD-L1 inhibitor, in combination with SoC chemoradiotherapy (SoC-CRT) against placebo (placebo plus SoC-CRT) as a first-line treatment in patients with locally advanced HNSCC being the primary endpoint PFS [92]. The study was completed recently, and the interim results presented in ESMO 2020 showed no significant improvement in PFS (based on 224 events) and OS (based on 131 events) with Avelumab plus SoC-CRT”. One reference was included in the reference list accordingly.

For IMvoke010 we have indicated that the efficacy is measured in terms of PFS. Furthermore, as this clinical trial focus on atezolizumab as adjuvant therapy, the sentence has been moved to the new subsection 4.3. ICB in the neoadjuvant setting. This new section was created to address a suggestion of one of the other reviewers. Thus, the current version in page 26, lines 537-539 reads as follows: In this regard, IMvoke010 (NCT03452137), an ongoing phase III clinical trial, aims to evaluate the efficacy in terms of PFS and the safety of Atezolizumab as adjuvant therapy in patients with high-risk locally advanced HNSCC [104].”

  • “Although no statistically significant differences in OS were found for the monotherapy or the combination versus SoC, higher survival, and response rates at 12 and 24 months demonstrated the clinical activity of durvalumab” (lines 403-405). This is a near-exact copy of the original EAGLE trial’s abstract conclusion: “There were no statistically significant differences in OS for durvalumab or durvalumab plus tremelimumab versus SoC. However, higher survival rates at 12 to 24 months and response rates demonstrate clinical activity for durvalumab”.

Response: We thank the reviewer for this insightful comment, we have rewriten the text and improved the description the results of the clinical trial. The corrected version is on page 11, lines 456-462.

  • “PD-L1 expression on tumor and immune cells has been associated with improved treatment outcomes in several cancers” (lines 522-523).

Response: We thank the reviewer for raising this point, we have changed the reference for one more update concerning the tumor types until April 2019 for which PD-L1 status has been approved by FDA as a predictive biomarker. We have also reformulated this section in order to address a comment of other reviewer regarding TPS and CPS in HNSCC and its usefulness in patient selection. The current version is on pages 27 and 28, lines 602-627. Six references were included in the reference list accordingly.

  • “Interestingly, intra-tumor heterogeneity is discussed as a potential biomarker of response to immunotherapy based on its impact on the immune profile of the TME and association with survival” (lines 529-531).

Response: We thank the reviewer for raising this point. We have deleted this sentence from the text as we consider it not suitable for this section. Particularly, in this section we discuss the intra-tumoral heterogeneity of PD-L1 expression and how it could impact on the TPS and CPS scores. Therefore, this sentence related to the genomic intra-tumoral heterogeneity, despite being currently associated and discussed as a potential biomarker of response to immunotherapy based on tis impact on the immune profile of the TME, is not related to PD-L1 expression as a biomarker and is associated with neo-antigens and TMB.

  • “In a scRNA-Seq study of primary tumors and lymph node metastasis biopsies from HNSCC patients, the immune types found varied in their proportions. However, they showed consistent expression profiles across patients.” (lines 562-564).

Response: We thank the reviewer for this comment. In the study of Puram and colleagues, when analyzing immune cell populations in HNSCC tumors by scRNA-Seq the authors found that there were no patient-specific subpopulations or cell states. Instead, all populations were present in all tumors and shared the same expression profiles across patients. What differentiated the TME between patients was the proportion in which different immune populations were present. They don’t report anything else about the differences in the composition of the immune infiltrate and don’t draw other conclusions related to this point, except for the exhaustion program found in a subpopulation of CD8+ T cells. To clarify this point we have reformulated our sentence related to these findings and added a further explanation of the implications for the observations made on the T-cell subpopulations, especially for the exhausted CD8+ T cells. Please see page 29, lines 659-670.